# Generalized symmetry enriched criticality in (3+1)d

**Benjamin Moy**

Department of Physics and Anthony J. Leggett Institute for Condensed Matter Theory,
University of Illinois Urbana-Champaign, Urbana, Illinois 61801, USA

## Abstract

We construct two classes of continuous phase transitions in 3+1 dimensions between gapped phases that break distinct generalized global symmetries. Our analysis focuses on $SU(N)/\mathbb{Z}_N$ gauge theory coupled to $N_f$ flavors of Majorana fermions in the adjoint representation. For $N$ even and sufficiently large odd $N_f$, upon imposing time-reversal symmetry and an $SO(N_f)$ flavor symmetry, the massless theory realizes a quantum critical point between a gapped phase in which a $\mathbb{Z}_N$ one-form symmetry is completely broken and a phase where it is broken to $\mathbb{Z}_2$, leading to $\mathbb{Z}_{N/2}$ topological order. We characterize the possible patterns of symmetry fractionalization in these phases and provide an explicit lattice model that exhibits the transition. The critical point has an enhanced symmetry, which includes non-invertible analogues of time-reversal symmetry. Enforcing a non-invertible time-reversal symmetry and the $SO(N_f)$ flavor symmetry, for $N$ and $N_f$ both odd, we demonstrate that this critical point can appear between a topologically ordered phase and a phase that spontaneously breaks the non-invertible time-reversal symmetry, furnishing an analogue of deconfined quantum criticality for generalized symmetries.

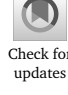
doi:10.21468/SciPostPhys.19.6.145

# 1   Introduction

The Landau theory of phases and phase transitions is a cornerstone of twentieth-century physics. In this framework, phases are distinguished by the global symmetries they preserve or spontaneously break, as detected by the expectation value of a local order parameter. Continuous transitions are associated with universal critical exponents that depend on the symmetries and number of dimensions of the system but not on its microscopic details.

Much of modern condensed matter physics is dedicated to phases and transitions that are not captured by Landau theory. Topologically ordered phases do not have any local order parameter but are characterized by long-range entanglement and excitations with fractional statistics [1, 2]. There also exist symmetry-protected topological phases (SPTs), which do not have topological order or symmetry breaking but nonetheless cannot be deformed into a trivial product state as long as a given symmetry is preserved [3–8]. These states are characterized by their response to background gauge fields or by the 't Hooft anomalies of their boundary states. Similarly, with a global symmetry, topological orders may be distinct as symmetry-enriched topological phases (SETs) [9, 10]. They may have a different response to background gauge fields that probe the symmetry or have distinct patterns of symmetry fractionalization—the excitations can carry different fractional quantum numbers of the symmetry.

While substantial progress has been made on topological phases themselves, a general theory of phase transitions between them is lacking. An important first step in developing such a theory is to identify examples of these continuous topological transitions. Some examples of continuous transitions between phases with distinct topological orders are well known in (2+1)d. For example, gauging the $\mathbb{Z}_2$ global symmetry of the (2+1)d Ising transition leads to the confinement-deconfinement transition of $\mathbb{Z}_2$ lattice gauge theory [1, 11]. However, in (3+1)d continuous transitions between phases with different topological orders remain elusive since most of these transitions turn out to be first order [12–15]. Although topological transitions may still arise from gauging the discrete symmetry of an ordinary symmetry-breaking

transition [16, 17], examples beyond these are rare. Therefore, it is of great importance to identify models with continuous transitions between phases with distinct topological orders in (3+1)d.

A possible source of insight on this problem is generalized global symmetries [18–26]. Two kinds of generalized symmetries will be important for us. The first is higher-form symmetry. An ordinary global symmetry acts on local operators through unitary (or antiunitary) symmetry operators that are topological and supported on codimension one manifolds. In contrast, the operators for a $q$-form global symmetry are topological but are defined on codimension $q + 1$ manifolds and act on operators supported on $q$-dimensional manifolds. Another generalization, non-invertible symmetry, has symmetry operators that are topological but have no inverse and thus are not unitary. An ordinary global symmetry is then said to be an invertible zero-form symmetry.

Generalized symmetries can be useful since this perspective allows us to draw analogies with conventional spontaneous symmetry breaking (SSB).[1] For example, Abelian topological orders may be viewed as spontaneously breaking an emergent discrete invertible one-form global symmetry.[2] In (2+1)d, the braiding of two Abelian anyons results in a phase factor, which can be interpreted as the action of a one-form symmetry associated with the worldline of one anyon acting on the other. Similarly, non-Abelian topological orders arise from the breaking of emergent non-invertible one-form symmetries.

Topological phase transitions can thus be viewed as part of a broader effort to understand transitions between phases that break different generalized global symmetries. This perspective has led to the development of an analogue of Ginzburg-Landau theory for invertible one-form symmetries known as mean string field theory (MSFT) [27]. While MSFT accurately captures phases and topological defects, it is difficult to use this framework to analyze phase transitions reliably, primarily because string observables have many more degrees of freedom than local operators.

In this work, we present two families of continuous phase transitions between phases that spontaneously break different generalized global symmetries. Our analysis focuses on $PSU(N) = SU(N)/\mathbb{Z}_N$ gauge theory coupled to $N_f$ odd flavors of Majorana fermions in the adjoint representation, a theory we refer to as $PSU(N)$ adjoint QCD. Imposing an $SO(N_f)$ flavor symmetry and time-reversal symmetry constrains the allowed deformations to a single relevant operator: a common mass term for all fermions. Tuning the mass $m$ can then induce a transition between distinct gapped phases. Similar constructions in $SU(N)$ gauge theory, which partially inspired this work, were found to host unconventional SPT transitions [28] and symmetry-breaking transitions [29, 30].

For sufficiently large $N_f$, when the fermions are massless, the theory will be either an interacting conformal field theory (CFT) [31, 32] or an infrared (IR) free theory, with the gauge coupling becoming small at low energies. The IR free theories are typically of limited interest to high energy theorists since their IR dynamics are simple by construction, and the gauge coupling becomes large in the ultraviolet (UV), rendering the field theory ill-defined at high enough energy scales. However, this issue is unimportant from the perspective of a low energy physicist, who regards the $PSU(N)$ adjoint QCD theory as an effective field theory valid near a phase transition, just as $\phi^4$ theory describes the (3+1)d Ising transition. Moreover, in the IR free case, since the gauge coupling becomes small in the IR, we can be sure that

---

[1]Of course there are also important differences. In realistic condensed matter systems, generalized symmetries are often either emergent or appear only in fine-tuned limits.

[2]To define SSB for a zero-form symmetry, we perturb the system with a local symmetry-breaking field $h(x)$ and examine whether the expectation value of the order parameter is nonzero if we first take the thermodynamic limit and then $h(x) \to 0$. This definition does not generalize to one-form symmetries since the observables are nonlocal. Instead, the appropriate criterion involves the expectation value of pairs of oppositely oriented loop operators wrapping nontrivial cycles; nonvanishing correlation at large separation indicates spontaneous symmetry breaking.

there is a direct continuous transition at $m = 0$ and can easily compute critical exponents reliably. For this reason, we focus on the IR free theories in this work, but we note that the topological transitions we observe may also occur for a theory with a smaller value of $N_f$ with an interacting CFT at the transition.

For $m > 0$, at energies far below the fermion mass, we can integrate out the fermions to obtain pure $PSU(N)$ gauge theory with vanishing theta angle. This phase has a $\mathbb{Z}_N$ magnetic one-form symmetry, which becomes spontaneously broken at low energies, leading to $\mathbb{Z}_N$ topological order (i.e., $\mathbb{Z}_N$ toric code topological order). The point-like anyons are magnetic monopoles of the $PSU(N)$ gauge theory, but since we view the $PSU(N)$ gauge theory as an emergent theory of a condensed matter system, we emphasize that these quasiparticles should not be confused with magnetic monopoles of the $U(1)$ electromagnetic field.

For $N$ even and $N_f$ odd, the $m < 0$ phase flows to $PSU(N)$ gauge theory with[3] $\theta = \pi N N_f$. As we discuss below, the theta term induces partial breaking of the $\mathbb{Z}_N$ one-form symmetry to $\mathbb{Z}_2$, resulting in $\mathbb{Z}_{N/2}$ topological order enriched by the $SO(N_f)$ flavor symmetry and time-reversal symmetry. For $N \geq 4$, this transition is between SETs with different topological orders. The $N = 2$ case, corresponding to gauge group $PSU(2) \cong SO(3)$, is a transition between a phase with $\mathbb{Z}_2$ topological order and an SPT state, providing a novel example of a continuous confinement-deconfinement transition. To our knowledge, this class of models is the first in (3+1)d with an exact one-form symmetry that have an unambiguously continuous transition between phases with different patterns of one-form symmetry breaking.

When the fermions are massless, the symmetry of $PSU(N)$ adjoint QCD is enhanced by a family of non-invertible time-reversal symmetries [33, 34], which are associated with topological operators $\mathsf{T}_n$ that act on the $PSU(N)$ theta angle as $\theta \to -\theta + 2\pi n$, where $n$ is an integer mod $N$. For $n \in N\mathbb{Z}$, this transformation reduces to the usual invertible time-reversal symmetry. This observation raises a natural question: What phases arise if we perturb the critical point by operators that preserve both the flavor symmetry and a non-invertible time-reversal symmetry $\mathsf{T}_n$?

Once again, the symmetries permit only a single relevant local operator that can tune a continuous transition between two gapped phases. An important non-perturbative constraint indicating a potentially rich phase diagram is that, for $N$ odd and $n > 0$ such that $\gcd(N, n) > 1$, the theory exhibits an anomaly: no single ground state can preserve both the $\mathbb{Z}_N$ magnetic one-form symmetry and the non-invertible time-reversal symmetry. Indeed, when these anomaly conditions are met and $N_f$ is odd, we show that the relevant deformation controls a transition between a phase with topological order and a phase that spontaneously breaks the non-invertible time-reversal symmetry. In the special case $n = 0$, the time-reversal symmetry is invertible, and the transition will be between a phase with $\mathbb{Z}_N$ topological order and a phase that breaks time-reversal symmetry spontaneously.

To illustrate this kind of transition in a class of models for which the time-reversal symmetry is non-invertible, we take $N = k^2$ and $n = 2k$ with odd $k > 1$ for concreteness. At low energies, where the fermions may be integrated out, one phase becomes pure $PSU(N)$ Yang-Mills at $\theta = \pi n$, and the other phase has $\theta = \pi(n + N N_f)$. The $\theta = 2\pi k$ phase preserves the non-invertible time-reversal symmetry but spontaneously breaks the $\mathbb{Z}_{k^2}$ magnetic one-form symmetry to $\mathbb{Z}_k$, leading to $\mathbb{Z}_k$ topological order. In contrast, the $\theta = \pi(2k + k^2 N_f)$ phase retains the full one-form symmetry but spontaneously breaks the non-invertible time-reversal symmetry, resulting in non-invertible domain walls that interpolate between distinct one-form SPT states. In both phases, the $SO(N_f)$ flavor symmetry remains unbroken. Since this transition is between two gapped phases that break completely different symmetries, it may be viewed as an analogue of deconfined quantum criticality [35–37] but for generalized symmetries.

---

[3]The normalization of the $PSU(N)$ theta term is such that $\theta$ has periodicity $2\pi N$.

Gapless phases and phase transitions with non-invertible symmetries have been explored in many recent works. In (1+1)d, several studies have examined transitions between phases with SSB of an ordinary symmetry and SSB of a non-invertible symmetry [38], transitions between phases that break different non-invertible symmetries [39], transitions between an SPT protected by a non-invertible symmetry [40] and a non-invertible SSB phase or another non-invertible SPT [41], transitions with Haagurup symmetry between Haagurup-symmetric gapped phases [42]. Gapless SPTs enriched by non-invertible symmetries have been investigated in several different dimensions [43–47]. Furthermore, recent efforts have sought to characterize (2+1)d conformal field theories (CFTs) arising at transitions between distinct topologically ordered phases enhanced by non-invertible one-form symmetries [48]. Our second construction introduces a novel example of a critical point in (3+1)d between a phase that spontaneously breaks an invertible one-form symmetry and a phase that spontaneously breaks a non-invertible symmetry, thereby broadening the known landscape of unconventional critical phenomena enriched by generalized symmetries.

We proceed as follows. To keep this work self-contained, we begin in Section 2 by reviewing the topological phases that occur in $SU(N)$ and $SU(N)/\mathbb{Z}_N$ gauge theories, and we discuss an SPT transition in $SU(N)$ adjoint QCD in Section 3. We then combine these ingredients in Section 4 to find a continuous topological transition in $SU(N)/\mathbb{Z}_N$ adjoint QCD, which in general is a transition between SETs that have different topological orders. In Section 5, we introduce the critical point between a topologically ordered phase and a phase that spontaneously breaks a non-invertible zero-form symmetry. We conclude in Section 6 with a discussion of our results and possible future directions. Additional details and technical background are included in the appendices.

## 2 Topological phases in pure Yang-Mills

Before we discuss non-Abelian gauge theories coupled to matter in the adjoint representation, we review the topological phases that occur in pure $SU(N)$ and $PSU(N) = SU(N)/\mathbb{Z}_N$ gauge theories since these phases will be realized at low energies in adjoint QCD when the fermions are massive. As we will explain below, pure $SU(N)$ gauge theory with a theta term is a $\mathbb{Z}_N$ one-form SPT at $\theta \in 2\pi\mathbb{Z}$. Correspondingly, $PSU(N)$ gauge theory at $\theta \in 2\pi\mathbb{Z}$ is characterized at low energies by a particular TQFT, which can describe a topologically ordered phase depending on the particular value of $\theta$.

### 2.1 $SU(N)$ gauge theory

We start by reviewing the physics of pure $SU(N)$ gauge theory in (3+1)d with a theta term. The action is

$$S_{\text{YM}}[\theta] = -\frac{1}{g^2}\int \text{Tr}(f \wedge \star f) + \frac{\theta}{8\pi^2}\int \text{Tr}(f \wedge f), \tag{1}$$

where $f = da - i\, a \wedge a$ is the field strength for the dynamical $SU(N)$ gauge field $a_\mu$ and the trace is taken in the fundamental representation. A Wilson loop represents the worldline of a probe electric quark-antiquark pair, and can be defined for any representation $\mathcal{R}$ of $SU(N)$ as

$$W_{\mathcal{R}}(\gamma) = \text{Tr}_{\mathcal{R}}\, \mathcal{P}\exp\left(i\oint_\gamma a\right), \tag{2}$$

where $\mathcal{P}$ denotes path ordering and $\gamma$ is a closed loop in spacetime. For the purposes of topological physics, the electric loop operators are organized into classes labeled the center of the gauge group. Thus, for $SU(N)$, which has a center of $\mathbb{Z}_N$, these classes are labeled

by an integer $q_e$ mod $N$ [49, 50]. For example, the fundamental representation has $q_e = 1$, but more generally, $q_e$ is the number of boxes (mod $N$) for the Young tableau diagram of the representation $\mathcal{R}$. The integer $q_e$ is defined mod $N$ because of screening; for instance, a Wilson loop in the adjoint representation can be opened and end on the field strength.

We can also introduce 't Hooft operators [51],

$$T_{q_m}(\gamma, \Omega), \tag{3}$$

where $\gamma$ is a loop attached to an open surface $\Omega$ such that $\gamma = \partial \Omega$. To insert an 't Hooft operator into the path integral, we transform the $SU(N)$ gauge field by

$$a_\mu \to U_\gamma \, a_\mu \left(U_\gamma\right)^{-1} + i \, U_\gamma \, \partial_\mu \left(U_\gamma\right)^{-1}, \tag{4}$$

where $U_\gamma$ is singular along $\gamma$. For any other closed curve $\gamma'$, parametrized by $s \in [0, 2\pi]$, that winds through $\gamma$ with winding number $w$ (in a specified direction), the singularity of $U_\gamma(s)$ is such that

$$U_\gamma(2\pi) = e^{2\pi i \, w/N} \, U_\gamma(0). \tag{5}$$

Despite the resemblance of Eq. (4) to a gauge transformation, the insertion of the 't Hooft operator is not a gauge transformation because of its singularity along $\gamma$. Physically, 't Hooft operators represent worldlines of magnetic probe particles characterized by an integer charge $q_m$ mod $N$ [49, 50, 52]. The surface $\Omega$ attached to $\gamma$ is the worldsheet of a Dirac string that is detectable by Wilson loops. Upon canonically quantizing the theory, at equal times the Wilson loop in the fundamental representation and the 't Hooft operator obey [51]

$$W_F(\gamma) \, T_{q_m}(\gamma', \Omega) = e^{2\pi i \, q_m \, \varphi(\gamma, \gamma')/N} \, T_{q_m}(\gamma', \Omega) \, W_F(\gamma), \tag{6}$$

where $\gamma$ and $\gamma' = \partial \Omega$ are loops in space and $\varphi(\gamma, \gamma')$ is their linking number. The Dirac string is thus detectable unless $q_m = 0$ mod $N$. Indeed, for a gauge theory with gauge group $G$, the 't Hooft operators that are genuine loop operators (i.e., do not require a choice of surface $\Omega$) are classified by the first homotopy group $\pi_1(G)$, which is trivial for $G = SU(N)$.

Combining Wilson loops and 't Hooft operators allows us to form a more general class of dyonic loop operators,

$$D_{(q_e, q_m)}(\gamma, \Omega) = W_{\mathcal{R}}(\gamma) \, T_{q_m}(\gamma, \Omega), \tag{7}$$

which carry both electric and magnetic charge. We denote their charges collectively as $(q_e, q_m)$. These operators are genuine loop operators in $SU(N)$ gauge theory only if $q_m = 0$ mod $N$.

The pure $SU(N)$ gauge theory has a global $\mathbb{Z}_N$ electric one-form symmetry, also known as the "center symmetry", that acts on Wilson loops. The symmetry operator is a surface operator $U_{q_m}(\Sigma)$, where $\Sigma$ is a closed surface and $q_m$ is an integer mod $N$. Physically, $U_{q_m}(\Sigma)$ can be thought of as the insertion of the worldsheet of a background magnetic flux tube. This operator may be constructed by taking the limit of the 't Hooft operator $T_{q_m}(\gamma, \Sigma)$ as $\gamma = \partial \Sigma$ is shrunk to a point so that $\Sigma$ is a closed surface. This surface operator acts on Wilson loops as

$$\langle U_{q_m}(\Sigma) \, W_{\mathcal{R}}(\gamma) \dots \rangle = e^{2\pi i \, q_e \, q_m \, \Phi(\gamma, \Sigma)/N} \langle W_{\mathcal{R}}(\gamma) \dots \rangle, \tag{8}$$

where $\Phi(\gamma, \Sigma)$ is the linking number in spacetime of the loop $\gamma$ and the closed surface $\Sigma$, and the ellipses denote insertions of other operators.

Pure $SU(N)$ gauge theory is believed to be gapped and non-degenerate at zero temperature for any $\theta \neq (2k+1)\pi$ with $k \in \mathbb{Z}$. Moreover, electric charges are confined, as signaled by the area law for a Wilson loop in the fundamental representation [53],

$$\langle W_F(\gamma) \rangle \sim e^{-\sigma \, \mathrm{Area}(\gamma)}, \tag{9}$$

where Area($\gamma$) is the area of the minimal surface that bounds the loop $\gamma$ and $\sigma$ is a constant called the string tension. The area law signals that the $\mathbb{Z}_N$ electric one-form symmetry is unbroken.

While the 't Hooft operator is not a genuine loop operator, it can still be used as a probe to characterize the phase of the gauge theory. At $\theta = 0$, where conventional confinement is expected, the basic 't Hooft operator (i.e., with $q_m = 1$) has a perimeter law [51],

$$\langle T_1(\gamma, \Omega) \rangle \sim e^{-\rho \, \text{Length}(\gamma)}, \tag{10}$$

where Length($\gamma$) is the length of the loop $\gamma$ and $\rho$ is a non-universal constant, which is generically scheme-dependent. This perimeter law is consistent with the notion of confinement arising from the condensation of magnetic monopoles [51, 54–56]. Specifically, because monopoles with charge $q_m = N$ are screened, we view this phase as arising from the condensation of charge $N$ monopoles.

The parameter $\theta$ is $2\pi$ periodic (on a spin manifold), so the spectrum of the theory at $\theta = 2\pi k$, where $k \in \mathbb{Z}$, is the same as at $\theta = 0$. However, the $\theta$ parameter gives an electric polarization charge to magnetic monopoles—a phenomenon known as the Witten effect [57]. After a change in theta by $\Delta\theta = 2\pi k$, a dyon that originally had charges $(q_e, q_m)$ becomes

$$(q_e, q_m) \rightarrow (q_e + k\, q_m, q_m). \tag{11}$$

As argued by 't Hooft, the Witten effect can lead to phases with condensed dyons known as oblique confinement [58, 59], which, as we discuss below, can lead to rich topological physics [18, 60–68]. Since the 't Hooft operators have a perimeter law at $\theta = 0$, a dyon with charges $(q_e, q_m)$ at $\theta = 0$ has a perimeter law at $\theta = 2\pi k$ if

$$(q_e + k\, q_m, q_m) = (0, q_m) \mod N, \tag{12}$$

so that these loops are purely magnetic once the Witten effect is taken into account. At $\theta = 2\pi k$, we then have

$$\langle D_{(-k\, q_m, q_m)}(\gamma, \Omega) \rangle \sim e^{-\tilde{\rho} \, \text{Length}(\gamma)}, \tag{13}$$

where we use the convention of labeling the dyon charges prior to the Witten effect (i.e., we label by the charges of dyons at $\theta = 0$). The dyon condensed at $\theta = 2\pi k$ likewise has charges $(-Nk, N)$. Here, the constant $\tilde{\rho}$ associated with the perimeter law is again non-universal and scheme-dependent.

Another important way to characterize an oblique confining phase is by its response upon coupling to a background field that probes the $\mathbb{Z}_N$ one-form symmetry. Following Ref. [69], we first embed the original $SU(N)$ gauge field $a_\mu$ into a $U(N)$ gauge field $\alpha_\mu$, which has field strength $(f_\alpha)_{\mu\nu}$. We then use a two-form Lagrange multiplier $\beta_{\mu\nu}$ to constrain $\text{Tr}(f_\alpha)$ to be trivial so that the theory is still $SU(N)$ gauge theory. Introducing a background $U(1)$ two-form gauge field $B_{\mu\nu}$ and a background $U(1)$ one-form gauge field $C_\mu$ such that $dC = NB$, we then couple these background fields so that the partition function is invariant under the gauge transformations,

$$B \rightarrow B + d\lambda, \qquad C \rightarrow C + N\lambda + d\xi, \qquad \alpha \rightarrow \alpha - \lambda \, \mathbb{I}_N, \tag{14}$$

where $\xi$ is a $2\pi$ periodic scalar field, $\lambda_\mu$ is a $U(1)$ one-form gauge field, and $\mathbb{I}_N$ is the $N$ by $N$

identity matrix. Upon coupling to these probes, the action is now

$$
\begin{aligned}
S_{\text{YM}}[\theta, B] = &-\frac{1}{g^2} \int \text{Tr}[(f_\alpha + B\,\mathbb{I}_N) \wedge \star(f_\alpha + B\,\mathbb{I}_N)] \\
&+ \frac{\theta}{8\pi^2} \int \text{Tr}[(f_\alpha + B\,\mathbb{I}_N) \wedge (f_\alpha + B\,\mathbb{I}_N)] \\
&- \frac{\theta}{8\pi^2} \int [\text{Tr}(f_\alpha) + NB] \wedge [\text{Tr}(f_\alpha) + NB] \\
&+ \frac{1}{2\pi} \int \beta \wedge [\text{Tr}(f_\alpha) + NB] + \frac{1}{2\pi} \int \tilde{\beta} \wedge (-\mathrm{d}C + NB),
\end{aligned}
\tag{15}
$$

where $\tilde{\beta}_{\mu\nu}$ is another two-form Lagrange multiplier that constrains $\mathrm{d}C = NB$. Without the background fields present, the $\theta$ angle is $2\pi$ periodic in the pure $SU(N)$ gauge theory because of the quantization of the theta term. If we take $\theta \to \theta + 2\pi k$, where $k \in \mathbb{Z}$, then the action changes (modulo an integer multiple of $2\pi$) by [69]

$$
S_{\text{SPT}}[B] = \frac{N(N-1)k}{4\pi} \int B \wedge B.
\tag{16}
$$

Consequently, assuming that $SU(N)$ gauge theory at $\theta = 0$ is a trivial confining phase, then the oblique confining state at $\theta = 2\pi k$ is an SPT protected by the $\mathbb{Z}_N$ electric one-form symmetry [18,70] and characterized by the response in Eq. (16). This SPT is nontrivial unless $k \in N\mathbb{Z}$ on a spin manifold or $k \in 2N\mathbb{Z}$ on a generic manifold.

The physical meaning of the response, Eq. (16), is related to dyon condensation. The background field $B_{\mu\nu}$ can be constructed from a configuration of the $\mathbb{Z}_N$ one-form symmetry operators $U_{q_m}(\Sigma)$ that end on loops. At $\theta = 0$, as discussed previously near Eq. (8), these operators are precisely the 't Hooft loops. At $\theta = 2\pi k$, the response, Eq. (16), indicates that the loops carry a charge $q_e = (N-1)k\,q_m = -k\,q_m \bmod N$ under the one-form symmetry. Thus, the loop at which the one-form symmetry operator $U_{q_m}(\Sigma)$ ends has charges $(-k\,q_m, q_m)$. As discussed near Eq. (13), dyons with these charges are precisely those that have a perimeter law at $\theta = 2\pi k$. Indeed, dyons with charges not of this form are confined and energetically suppressed, so the symmetry operators can only end on loops that have a perimeter law. We recall that the dyonic loops that have a perimeter law must be attached to surfaces (unless $k \in N\mathbb{Z}$, in which case the SPT is trivial), so these objects are analogous to string order parameters in (1+1)d zero-form SPTs [71].

Because there is a change in response to background fields as $\theta$ is tuned continuously from $\theta = 2\pi k$ to $\theta = 2\pi(k+1)$, there must be at least one phase transition as $\theta$ is varied. Based on evidence from lattice simulations [72–74], large $N$ [75], 't Hooft anomalies [69], and deformations of supersymmetric theories [76], a single first order transition with spontaneously broken time-reversal symmetry [77] is believed to occur at $\theta = 2\pi k + \pi$, where the $\mathbb{Z}_N$ one-form SPT states become degenerate. Although this picture is not fully settled, it holds rigorously for large enough $N$ [75].

To summarize, pure $SU(N)$ Yang-Mills theory has a $\mathbb{Z}_N$ electric one-form symmetry and realizes SPT phases protected by this symmetry at $\theta \in 2\pi\mathbb{Z}$. These SPT phases are realized by the mechanism of dyon condensation and oblique confinement.

## 2.2 $SU(N)/\mathbb{Z}_N$ gauge theory

We next review the continuum description of $PSU(N) = SU(N)/\mathbb{Z}_N$ Yang-Mills theory [18,50, 78,79], which includes $PSU(2) \cong SO(3)$ as a special case. The $PSU(N)$ gauge theory can be constructed by gauging the $\mathbb{Z}_N$ electric one-form symmetry of $SU(N)$ gauge theory. Hence, we

promote the background fields in Eq. (15) to dynamical fields. The explicit form of the action is[4]

$$S_{PSU(N)}[\theta, B] = S_{\text{YM}}[\theta, b] + \frac{N}{2\pi} \int b \wedge B, \tag{17}$$

where $S_{\text{YM}}[\theta, b]$ is the same as Eq. (15) but with $C_\mu \to c_\mu$ and $B_{\mu\nu} \to b_{\mu\nu}$ promoted to dynamical $U(1)$ one-form and two-form gauge fields respectively.[5] Naturally, the fields $\alpha_\mu$, $c_\mu$, and $b_{\mu\nu}$ have gauge transformations descending from Eq. (14). We have also introduced a new $\mathbb{Z}_N$ two-form background gauge field $B_{\mu\nu}$, which probes a $\mathbb{Z}_N$ magnetic one-form symmetry.

We constructed $PSU(N)$ gauge theory by gauging the full electric one-form symmetry of $SU(N)$ gauge theory, so unsurprisingly, the center of $PSU(N)$ is trivial, and $PSU(N)$ gauge theory accordingly has no electric one-form symmetry. Because of the coupling to the dynamical two-form gauge field $b_{\mu\nu}$, a generic Wilson loop is no longer gauge invariant. Rather, it is supported on a loop $\gamma = \partial\Omega$ attached to an open surface $\Omega$ as

$$\mathcal{W}_\mathcal{R}(\gamma, \Omega) = \left[ \text{Tr}_\mathcal{R} \, \mathcal{P} \exp\left( i \oint_\gamma \alpha \right) \right] \exp\left( i q_e \int_\Omega b \right), \tag{18}$$

where $q_e$ is the one-form gauge charge for the representation $\mathcal{R}$. The attached surface is trivial only if $q_e \in N\mathbb{Z}$, which is true for the adjoint representation, but a Wilson loop in the adjoint representation is screened by the gluons.

The possible magnetic charges are classified by $\pi_1(PSU(N)) \cong \mathbb{Z}_N$, so there should be $N$ distinct 't Hooft loops that are genuine loop operators. These 't Hooft loops are expressed most easily if we integrate out $c_\mu$, which implements the constraint that $\tilde{\beta} = \mathrm{d}\tilde{a}$, where $\tilde{a}_\mu$ is a $U(1)$ one-form gauge field. The 't Hooft loops are

$$\mathcal{T}(\gamma)^{q_m} = \exp\left( i q_m \oint_\gamma \tilde{a} \right), \tag{19}$$

where $q_m$ is an integer mod $N$. The 't Hooft loop is now always a genuine loop operator. Indeed, by definition, gauging the electric one-form symmetry of $SU(N)$ gauge theory trivializes the symmetry operator $U_{q_m}(\Sigma)$. Because 't Hooft loops are attached to open surface versions of $U_{q_m}(\Sigma)$, gauging the electric one-form symmetry renders the attached surface trivial, turning the 't Hooft loops into genuine loop operators.

As with $SU(N)$ gauge theory, dyon operators can be formed from products of the Wilson and 't Hooft loops as

$$\mathcal{D}_{(q_e, q_m)}(\gamma, \Omega) = \mathcal{W}_\mathcal{R}(\gamma, \Omega) \mathcal{T}(\gamma)^{q_m}. \tag{20}$$

We again label the dyonic operators by their charges $(q_e, q_m)$ at $\theta = 0$, prior to the Witten effect. These operators are genuine loop operators in $PSU(N)$ gauge theory only if $q_e = 0$ mod $N$.

A magnetic $\mathbb{Z}_N$ one-form global symmetry, probed by the $\mathbb{Z}_N$ two-form background field $B_{\mu\nu}$, acts on the 't Hooft loops. The operator, supported on a closed surface $\Sigma$, that acts with this symmetry is

$$\mathcal{U}(\Sigma) = \exp\left( -i \oint_\Sigma b \right). \tag{21}$$

---

[4]We could have added a discrete theta term [50] (proportional to the integral of $b \wedge b$) to the action, but this term can be absorbed into the definition of $\theta$.

[5]The two-form gauge field $b_{\mu\nu}$ is roughly $b = (2\pi/N) w_2^{PSU(N)}$, where $w_2^{PSU(N)}$ is the second Stiefel-Whitney class for $PSU(N)$, which characterizes the obstruction to lift the $PSU(N)$ gauge bundle to an $SU(N)$ bundle.

This operator is a closed surface version of the Wilson loop, Eq. (18), so it can be viewed as the worldsheet of a string of electric flux.[6] This magnetic one-form symmetry is the dual symmetry that arises after gauging the electric one-form symmetry.

Next, we characterize the phases realized in $PSU(N)$ gauge theory. Gauging the $\mathbb{Z}_N$ electric one-form symmetry does not modify whether operators have a perimeter law or area law. Therefore, at $\theta = 0$ the basic 't Hooft loop has a perimeter law, signaling deconfinement of monopoles, and the charge $N$ monopoles are condensed. But in $PSU(N)$ gauge theory, since the 't Hooft loops are genuine loop operators, their deconfinement signals that the $\mathbb{Z}_N$ magnetic one-form symmetry is spontaneously broken at low energies, resulting in topological order. Indeed, assuming that $SU(N)$ gauge theory at $\theta = 0$ is a trivial confining phase, then gauging its $\mathbb{Z}_N$ electric one-form symmetry results in a phase with a TQFT of

$$S_{\text{BF}}[B] = \frac{N}{2\pi} \int b \wedge (\mathrm{d}\tilde{a} + B), \tag{22}$$

which is BF theory at level $N$. This relation between the phases of $SU(N)$ gauge theory and $PSU(N)$ gauge theory has an analogue in (3+1)d $\mathbb{Z}_N$ lattice gauge theory, which has a $\mathbb{Z}_N$ electric one-form symmetry. Under gauging this $\mathbb{Z}_N$ one-form symmetry and modifying the gauge coupling in a particular way, the partition function for $\mathbb{Z}_N$ lattice gauge theory remains the same, but the confining phase, which is topologically trivial, is exchanged with the topologically ordered deconfined phase [78, 80, 81].

Following the discussion from Section 2.1, shifting $\theta \to \theta + 2\pi k$, where $k \in \mathbb{Z}$, changes the action by Eq. (16) but now with $B_{\mu\nu} \to b_{\mu\nu}$ dynamical. Consequently, the periodicity of $\theta$ in $PSU(N)$ gauge theory is $2\pi N$. At $\theta = 2\pi k$, the condensed dyons have charges $(-Nk, N)$, and the dyons obeying a perimeter law have charges of the form

$$(q_e, q_m) = (-k q_m, q_m), \tag{23}$$

which is consistent with the $2\pi N$ periodicity of $\theta$ since $q_e$ and $q_m$ are integers defined mod $N$. However, not all of these dyons in Eq. (23) correspond to genuine loop operators. The one-form symmetry can act nontrivially only on operators supported on noncontractible loops, but loops attached to a surface are always contractible.

The subset of dyons in Eq. (23) associated with genuine loop operators must have electric charges satisfying $q_e = -k q_m = 0 \bmod N$. Such operators must thus have magnetic charges such that $q_m \in \frac{N}{L}\mathbb{Z}$, where $L = \gcd(N, k)$. The dyonic loop operators that have a perimeter law and are genuine loop operators therefore have charges

$$(q_e, q_m) = \left(-\frac{Nk}{L} q, \frac{N}{L} q\right) = \left(0, \frac{N}{L} q\right) \quad \bmod N, \tag{24}$$

where $q$ is an integer mod $L$. The minimal magnetic charge of these deconfined dyons is $N/L$, so the $\mathbb{Z}_N$ magnetic one-form symmetry is spontaneously broken to $\mathbb{Z}_{N/L}$ in this phase. This phase is characterized by a topological order that is the same as in the deconfined phase of $\mathbb{Z}_N/\mathbb{Z}_{N/L} \cong \mathbb{Z}_L$ gauge theory. Indeed, the braiding phase between a deconfined magnetic charge $\mathcal{T}(\gamma)^{Nq/L}$ and an electric flux tube $\mathcal{U}(\Sigma)^{q'}$ is

$$\langle \mathcal{T}(\gamma)^{Nq/L} \mathcal{U}(\Sigma)^{q'} \rangle = \exp\left(\frac{2\pi i}{L} q q' \Phi(\gamma, \Sigma)\right), \tag{25}$$

where $\Phi(\gamma, \Sigma)$ is the linking number of $\gamma$ and $\Sigma$ in (3+1)d spacetime.

---

[6]The origin of the minus sign in the exponent of Eq. (21) can be traced to the ordering of operators in Eq. (6).

Naturally, the effective TQFT that describes the phase at $\theta = 2\pi k$ is the gauged version of the SPT action, Eq. (16). The specific action for the TQFT is

$$S_{\text{TQFT}}[p, \tilde{a}, b, B] = \frac{N}{2\pi} \int b \wedge (d\tilde{a} + B) + \frac{Np}{4\pi} \int b \wedge b \,, \tag{26}$$

where $p = (N-1)k$. As reviewed in Appendix A, this TQFT indeed describes a topological order realized by spontaneously breaking a $\mathbb{Z}_N$ one-form global symmetry to $\mathbb{Z}_{N/L}$ at low energies, where $L = \gcd(N, p) = \gcd(N, k)$. Phases of this type are also realized [65, 82] in a lattice model introduced by Cardy and Rabinovici [59, 83].

The unbroken $\mathbb{Z}_{N/L}$ one-form symmetry also has a nontrivial response to a background field $B_{\mu\nu}$, given by

$$S_{\text{resp}}[B] = \frac{Nr}{4\pi L} \int B \wedge B \,, \tag{27}$$

where $r$ is an integer mod $N/L$ such that $rp = r(N-1)k = -L \mod N$. The interpretation of this response is similar to that of Eq. (16). A surface operator that acts with the unbroken $\mathbb{Z}_{N/L}$ magnetic one-form symmetry is $\mathcal{U}(\Sigma)^L$. The configurations of the background field $B_{\mu\nu}$ are constructed by allowing these surface operators to end on loops. Because $\mathcal{U}(\Sigma)^L$ represents an electric flux tube, it must terminate on a loop with electric charge $q_e = -L$. This loop must have a perimeter law, so its charges must be of the form in Eq. (23) so that

$$q_e = -k q_m = -L \mod N \,. \tag{28}$$

A solution to this equation for the magnetic charge is then $q_m = r \mod N$ so that the loop on which $\mathcal{U}(\Sigma)^L$ ends has charges $(-L, r)$. Likewise, the symmetry operator $\left[\mathcal{U}(\Sigma)^L\right]^{Mk/L}$, where $M$ is an integer mod $N/L$, must end on a loop with charges

$$(q_e, q_m) = (-kM, M) \,. \tag{29}$$

These dyons are those that have a perimeter law but are not genuine line operators. Thus, although these operators do not contribute to the topological order, they can have physical consequences that manifest in the response, Eq. (27).

Similar to $SU(N)$ gauge theory, as $\theta$ is varied from $\theta = 2\pi k$ to $\theta = 2\pi(k+1)$, there must be a phase transition. It is believed that a first order transition occurs at $\theta = 2\pi k + \pi$, where the ground states of the TQFT, Eq. (26), with $p = (N-1)k$ are degnerate with the ground states of the same TQFT but with $p = (N-1)(k+1)$. This statement is rigorous for large $N$ [75]. If $\gcd(N, k) = \gcd(N, k+1) = 1$, then there are two degenerate $\mathbb{Z}_N$ one-form SPTs at the first order transition. Otherwise, there will be topologically ordered ground states degenerate with an SPT state.

## 3 $SU(N)$ adjoint QCD: SPT transition

Now that we have established the gapped phases of pure $SU(N)$ and $PSU(N)$ gauge theories, we are equipped to couple to matter and examine the transitions between these topological phases. We begin with $SU(N)$ gauge theory coupled to $N_f$ flavors of Majorana fermions in the adjoint representation, which we refer to as $SU(N)$ adjoint QCD. We focus on the case in which $N$ is even and $N_f$ is odd. Because the adjoint representation is real, coupling to the $SU(N)$ gauge field is consistent with imposing the Majorana condition. The relation of this theory to SPT transitions was previously studied in Ref. [28], but in that case, the authors explicitly broke the electric one-form symmetry and found transitions between distinct zero-form SPTs. In this work, because we are ultimately interested in $SU(N)/\mathbb{Z}_N$ gauge theory

coupled to fermions, we will leave the electric one-form symmetry unbroken in anticipation of Section 4 where we will gauge this symmetry.

Since the fermions are in the adjoint representation of $SU(N)$, it is convenient to express the fermions as matrices, given by[7] $\psi_J(x) = \psi_J^\alpha(x)\,T^\alpha$, where $J$ is the flavor index, $\alpha$ is the gauge index, and $T^\alpha$ are the generators of the $\mathfrak{su}(N)$ Lie algebra in the fundamental representation. The action is

$$S_{\text{AQCD}} = \int d^4x \sum_{J=1}^{N_f} \text{Tr}\left[\psi_J^T(x)\,\mathcal{C}\,(i\slashed{D}_a - m)\,\psi_J(x)\right] - \frac{1}{g^2}\int \text{Tr}(f \wedge \star f), \qquad (30)$$

where we define $\slashed{D}_a \psi_J = \gamma^\mu\left(\partial_\mu \psi_J - i[a_\mu, \psi_J]\right)$, $\psi_J^T(x)$ denotes the transpose of $\psi_J(x)$ for spinor indices only, $\gamma^\mu$ are Dirac matrices, and $m$ is the fermion mass. The charge conjugation matrix $\mathcal{C}$ is a unitary matrix acting on spinor indices that obeys $\mathcal{C}\gamma^\mu\mathcal{C}^{-1} = -(\gamma^\mu)^T$ and $\mathcal{C}^T = -\mathcal{C}$. The $\psi_J(x)$ are Majorana fermions obeying the constraint, $\bar\psi_J(x) = \psi_J^T(x)\mathcal{C}$. A brief review of Majorana fermions may be found in Appendix B.

### 3.1 Global symmetries

We proceed by describing the global symmetries of $SU(N)$ adjoint QCD, Eq. (30). Since the fermionic matter is coupled to the gauge field in the adjoint representation, the $\mathbb{Z}_N$ electric one-form symmetry of the pure $SU(N)$ gauge theory is retained. The zero-form symmetry that will be important for us is

$$G = SO(N_f) \times \mathbb{Z}_2^T, \qquad (31)$$

where $\mathbb{Z}_2^T$ is time-reversal symmetry and $SO(N_f)$ is the global flavor symmetry[8] of the fermions at a generic mass $m$, acting as

$$\psi_J(x) \to \mathcal{O}_{JK}\,\psi_K(x), \qquad (32)$$

for some matrix $\mathcal{O} \in SO(N_f)$. Time-reversal symmetry is associated with an antiunitary operator $\mathsf{T}$ that acts on the fields as

$$\psi_J^\alpha(\mathbf{x}, t) \to \mathcal{C}\gamma^5\,\psi_J^\alpha(\mathbf{x}, -t), \qquad a_0^\alpha(\mathbf{x}, t) \to -a_0^\alpha(\mathbf{x}, -t), \qquad a_j^\alpha(\mathbf{x}, t) \to a_j^\alpha(\mathbf{x}, -t). \qquad (33)$$

Hence, electric charge is odd under time-reversal while magnetic charge is even. From the general properties of $\mathcal{C}$ described above, we can also deduce that $\mathsf{T}^2 = (-1)^F$, where $(-1)^F$ is fermion parity. Since fermion parity acts as a global symmetry, adjoint QCD describes a fermionic theory in the sense that there exist local gauge invariant fermionic operators, such as $\text{Tr}[(\psi_J^T\,\mathcal{C}\,\psi_J)\psi_K]$. An important observation is that every gauge invariant bosonic local operator is a Kramers singlet, whereas every gauge invariant fermionic local operator transforms under time-reversal as a Kramers doublet. We refer to this constraint as a spin/Kramers relation (in analogy with a spin/charge relation [84]). A formal consequence is that the Wick rotated theory (in Euclidean spacetime) may be placed on a non-orientable manifold that admits a Pin$^+$ structure [85]. A manifold of this type must have a trivial second Stiefel-Whitney class, $w_2 = 0$.

Parity and charge conjugation symmetries are not essential for our purposes in this work, but we mention them for completeness. Parity is associated with a unitary operator $\mathsf{P}$ that acts on the fields as

$$\psi_J^\alpha(\mathbf{x}, t) \to i\gamma^0\,\psi_J^\alpha(-\mathbf{x}, t), \qquad a_0^\alpha(\mathbf{x}, t) \to a_0^\alpha(-\mathbf{x}, t), \qquad a_j^\alpha(\mathbf{x}, t) \to -a_j^\alpha(-\mathbf{x}, t). \qquad (34)$$

---

[7] We suppress spinor indices for simplicity.

[8] For $m \neq 0$, the full flavor symmetry is $O(N_f)$. For odd $N_f$, this group factorizes to $O(N_f) = SO(N_f) \times \mathbb{Z}_2^F$ (it is a semidirect product for even $N_f$), but the $\mathbb{Z}_2^F$ factor is equivalent to fermion parity. As we discuss below, this symmetry is already included in time-reversal.

Acting with parity twice then also gives $\mathsf{P}^2 = (-1)^F$. For $N > 2$, there is a global charge conjugation symmetry, which is associated with a unitary operator $\mathsf{C}$ acting as

$$\begin{aligned}
\psi_J(\mathbf{x}, t) &= \psi_J^\alpha(\mathbf{x}, t) T^\alpha \rightarrow -\psi_J^\alpha(\mathbf{x}, t)(T^\alpha)^* \,, \\
a_\mu(\mathbf{x}, t) &= a_\mu^\alpha(\mathbf{x}, t) T^\alpha \rightarrow -a_\mu^\alpha(\mathbf{x}, t)(T^\alpha)^* \,,
\end{aligned} \tag{35}$$

but for $N = 2$ this transformation is equivalent to a gauge transformation. These discrete symmetries are not entirely independent because of the CPT theorem.

At the massless point, $m = 0$, the zero-form flavor symmetry is enhanced to

$$G_{SU(N)} = \frac{SU(N_f) \times \mathbb{Z}_{2NN_f}}{\mathbb{Z}_{N_f}} \,. \tag{36}$$

The $\mathbb{Z}_{2NN_f}$ factor is a remnant of the classical $U(1)$ axial symmetry,

$$\psi_J(x) \rightarrow e^{i\vartheta\gamma^5} \psi_J(x). \tag{37}$$

By the Adler-Bell-Jackiw anomaly [86, 87], this transformation is equivalent to a shift of the $SU(N)$ theta angle by

$$\theta \rightarrow \theta + 2NN_f\,\vartheta. \tag{38}$$

Since the $SU(N)$ theta angle has periodicity $2\pi$, the action is invariant mod $2\pi$ under the transformation, Eq. (37), only if $NN_f\,\vartheta \in \pi\mathbb{Z}$. The axial global symmetry is thus anomalously broken to $\mathbb{Z}_{2NN_f}$ quantum mechanically.

This $\mathbb{Z}_{2NN_f}$ symmetry has a mixed 't Hooft anomalies. Suppose we couple Eq. (30) to a background $\mathbb{Z}_N$ two-form gauge field $B_{\mu\nu}$, a background gauge field $A_\mu$ for the $SO(N_f)$ flavor symmetry,[9] and a background metric $g_{\mu\nu}$ (to keep track of thermal response [88–91]). Transforming $\psi_J(x)$ by the discrete axial transformation,

$$\psi_J(x) \rightarrow e^{i\frac{2\pi k}{2NN_f}\gamma^5} \psi_J(x), \tag{39}$$

for some integer $k$, is equivalent to changing the action by

$$\Delta S = \frac{N(N-1)k}{4\pi} \int B \wedge B + \frac{\frac{2\pi k}{NN_f}(N^2 - 1)}{8\pi^2} \int \mathrm{Tr}(F \wedge F) + \frac{\frac{2\pi k}{N}(N^2 - 1)}{384\pi^2} \int \mathrm{Tr}(R \wedge R), \tag{40}$$

where $F_{\mu\nu}$ is the two-form field strength of the $SO(N_f)$ background field $A_\mu$ and $R$ is the curvature two-form.[10] Eq. (40) signals a mixed 't Hooft anomaly for the $\mathbb{Z}_{2NN_f}$ axial symmetry with the $\mathbb{Z}_N$ one-form symmetry, the flavor symmetry, and gravity. The first term in Eq. (40) arises because Eq. (39) is equivalent to a change in the $SU(N)$ theta angle by $2\pi k$ (cf. Eq. (38)). As discussed in Section 2.1, upon coupling to a background two-form gauge field $B_{\mu\nu}$, a shift in the $\theta$ angle of $SU(N)$ gauge theory by $2\pi k$ is equivalent to stacking with an SPT, Eq. (16), which matches the first term of Eq. (40).

## 3.2 Massive phases: SPTs

Next, we discuss the phases that occur when the fermions are massive. At low energies, well below the mass scale $|m|$, the fermions may be integrated out. For $m > 0$, the physics is governed by pure $SU(N)$ gauge theory with $\theta = 0$, which is gapped, confining, and topologically trivial. For $m < 0$, integrating out the fermions gives pure $SU(N)$ gauge theory with

---

[9]We couple only to a background field for the $SO(N_f)$ subgroup rather than $SU(N_f)$ because this subgroup is the symmetry that remains when we introduce masses for the fermions.

[10]See Appendix C for a more detailed definition of $R$ and a review of how to couple fermions to gravity in (3+1)d.

$\theta = \pi N N_f$, which is an integer multiple of $2\pi$ for $N$ even. As discussed in Section 2.1, $SU(N)$ gauge theory is also gapped and topologically trivial at this value of $\theta$, but it can differ from the $\theta = 0$ state as an SPT.

To understand precisely what SPT this state is, we again couple to a background metric $g_{\mu\nu}$, a background field $A_\mu$ for the $SO(N_f)$ flavor symmetry, and a background field $B_{\mu\nu}$ for the $\mathbb{Z}_N$ electric one-form symmetry. The theory with negative mass $m = -|m|$ is equivalent to the theory with a positive mass $m = |m|$ acted upon by an axial transformation, Eq. (39), with $k = N N_f/2$. Thus, if we choose a regularization such that the positive mass phase is a trivial SPT, then the negative mass phase is a nontrivial SPT with classical action,

$$S_{\text{SPT}} = \frac{N^2(N-1)N_f}{8\pi} \int B \wedge B + \frac{\pi(N^2-1)}{8\pi^2} \int \text{Tr}(F \wedge F) + \frac{\pi N_f(N^2-1)}{384\pi^2} \int \text{Tr}(R \wedge R). \quad (41)$$

The negative mass and positive mass phases thus differ as fermionic SPTs protected by the $\mathbb{Z}_N$ electric one-form symmetry, the $SO(N_f)$ flavor symmetry, and time-reversal symmetry.

If we introduce a (2+1)d boundary to the SPT state in Eq. (41), several different boundary states are possible. The bulk anomaly from the second and third terms of Eq. (41) can be matched at the boundary by $N_f(N^2-1)$ free massless Majorana fermions. The anomaly for the one-form symmetry can be matched by $SU(N)$ Chern-Simons theory with level $N N_f/2$, though other boundary states are also possible [69, 79]. A simple way to check that this topological order matches the anomaly is to observe that the bulk theory is $SU(N)$ gauge theory at $\theta = \pi N N_f$, which can be coupled to the two-form gauge field in a manifestly gauge invariant way as in Section 2.1. Applying Stokes' theorem to the $SU(N)$ theta term yields $SU(N)$ Chern-Simons theory at the boundary with level $N N_f/2$.

## 3.3 Phase transition

Having established the SPT phases that arise for $m \neq 0$, we address the transition between these phases, which is related to the IR fate of the massless point, $m = 0$. The low energy dynamics crucially depends on the number of flavors $N_f$. If the non-Abelian gauge field is coupled to $N_f(\mathcal{R})$ flavors of massless Majorana fermions in representation $\mathcal{R}$ is, then under the scale change $x_\mu \to e^\ell x_\mu$, the one-loop beta function[11] for the Yang-Mills coupling $g$ is [92, 93]

$$\beta(g) = \frac{dg}{d\ell} = (c_{\text{YM}} - c_{\text{M}}) g^3, \qquad c_{\text{M}} = \frac{2}{3(4\pi)^2} \sum_{\mathcal{R}} I_{\mathcal{R}} N_f(\mathcal{R}), \qquad c_{\text{YM}} = \frac{11}{3(4\pi)^2} I_{\text{adj}}, \quad (42)$$

where the group theoretic factor $I_{\mathcal{R}}$ is given by $\text{Tr}(T^\alpha_{\mathcal{R}} T^\beta_{\mathcal{R}}) = I_{\mathcal{R}} \delta^{\alpha\beta}$ for generators $T^\alpha_{\mathcal{R}}$ of the gauge group in representation $\mathcal{R}$. For $SU(N)$, we have $I_f = 1/2$ for the fundamental representation and $I_{\text{adj}} = N$ for the adjoint representation.

For $N_f$ flavors of Majorana fermions in the adjoint representation, the beta function in Eq. (42) indicates that the $m = 0$ point of $SU(N)$ adjoint QCD, Eq. (30), is infrared free for $N_f \geq 6$. As discussed in our introduction, since the gauge coupling becomes small in the IR, we can be sure that there is a direct continuous transition at $m = 0$ from the trivial SPT to the nontrivial SPT (cf. Eq. (41)), as depicted in Figure 1. The critical exponents can also be easily determined reliably in this case. Because we are interested in $N_f$ odd, the smallest value for which the transition is unambiguously continuous is $N_f = 7$.

---

[11] Our sign convention is such that $\beta(g) > 0$ signals a flow to strong coupling at low energies.



$$\underset{\underset{m < 0}{G \times \mathbb{Z}_N^{(1)} \text{ SPT}}}{\rule{5cm}{0pt}} \bullet \underset{\underset{m > 0}{\text{Trivial}}}{\rule{5cm}{0pt}}$$

Figure 1: Schematic depiction of the phase diagram of $SU(N)$ adjoint QCD, Eq. (30), for $N$ even and $N_f$ odd flavors of Majorana fermions, as a function of the mass $m$. For $m > 0$, the theory flows to $SU(N)$ gauge theory at $\theta = 0$, which is adiabatically connected to a trivial product state. For $m < 0$, the state is a nontrivial fermionic SPT protected by the $\mathbb{Z}_N^{(1)}$ electric one-form symmetry and $G = SO(N_f) \times \mathbb{Z}_2^T$, which consists of a flavor symmetry and time-reversal.

The cases $2 \le N_f \le 5$ are less well understood,[12] but we briefly summarize what is known about them here for completeness. The standard lore (see Ref. [96], for example) is that higher values of $N_f$ are interacting conformal field theories of the Banks-Zaks type [31, 32] while smaller values of $N_f$ ultimately confine and spontaneously break $G_{SU(N)} \to O(N_f)$, resulting in $N$ copies of an $SU(N_f)/SO(N_f)$ sigma model. But this picture is not firmly established, so other infrared behaviors are possible [97, 98] as long as they are consistent with anomaly constraints [99]. Current evidence suggests that $N_f = 4$ and $N_f = 5$ are conformal [100–107]. Thus, the $N_f = 5$ theory may also have a continuous SPT transition associated with an interacting CFT.[13]

The $N_f = 1$ theory is better understood since it has supersymmetry. This theory confines and breaks $G_{SU(N)} = \mathbb{Z}_{2N} \to \mathbb{Z}_2^F$ spontaneously [108, 109], resulting in $N$ degenerate ground states. Each of these $N$ vacua is associated with the condensation of a different dyon [110–112] and a $\mathbb{Z}_N$ one-form SPT of the form in Eq. (16) [18]. In this case, the transition between the $m > 0$ phase and $m < 0$ phase is first order since two of the $N$ ground states at the $m = 0$ point are associated with the trivial state at $m > 0$ and the nontrivial SPT at $m < 0$.

Hence, the global flavor symmetry $G = SO(N_f) \times \mathbb{Z}_2^T$ plays an important role. At the $m = 0$ point, the mass term in Eq. (30) is the only relevant perturbation that respects the symmetry $G$. For large enough $N_f$, if this symmetry is preserved, then there is a continuous transition directly between the two SPT states as in Figure 1. If the symmetry $G$ is not mandated, then other perturbations can drive the $m = 0$ theory to various intermediate phases or to a first-order transition.

# 4 $SU(N)/\mathbb{Z}_N$ adjoint QCD: SET transition

We now turn to our first main result—a topological transition between different (3+1)d SETs that have distinct patterns of one-form symmetry breaking (i.e., different topological orders). The theory in which this transition occurs is $SU(N)/\mathbb{Z}_N = PSU(N)$ gauge theory coupled to $N_f$ flavors of Majorana fermions in the adjoint representation, where we again take $N$ even

---

[12]CFTs in 4d have a universal quantity $a$ that is similar to the central charge of a 2d CFT [94]. For the IR free theories, we can compute that $a = (62 + N_f/2)(N^2 - 1)$, where we use units in which $a = 1$ for a single real free massless scalar field. For the asymptotically free theories, which have $N_f \le 5$, the UV value of $a$ is also $a_{UV} = (62 + N_f/2)(N^2 - 1)$. Since $a$ monotonically decreases along RG flows [95], a useful constraint is that the IR value for $N_f \le 5$ must satisfy $a_{IR} < a_{UV}$.

[13]There does not necessarily have to be a direct transition between the two SPT phases in this case. It is possible that there could be an intermediate "quantum phase" that is not obvious from the Lagrangian description, or criticality could possibly extend to small nonzero $m$.

and $N_f$ odd. The action for $PSU(N)$ adjoint QCD theory is

$$
\begin{aligned}
S_0 = \int \mathrm{d}^4 x \sum_{J=1}^{N_f} \mathrm{Tr}\big[\psi_J^T(x)\, C\, (i\slashed{D}_{\alpha+c\,\mathbb{I}_N} - m)\,\psi_J(x)\big] \\
- \frac{1}{g^2}\int \mathrm{Tr}[(f_\alpha + b\,\mathbb{I}_N)\wedge \star(f_\alpha + b\,\mathbb{I}_N)] + \frac{1}{2\pi}\int \beta \wedge [\mathrm{Tr}(f_\alpha) + \mathrm{d}c] \\
+ \frac{1}{2\pi}\int \tilde{\beta}\wedge(-\mathrm{d}c + Nb) + \frac{N}{2\pi}\int b\wedge B,
\end{aligned}
\tag{43}
$$

where we have gauged the $\mathbb{Z}_N$ one-form symmetry of Eq. (30) using the same methods as in Section 2.2. Namely, $\alpha_\mu$ is a dynamical $U(N)$ gauge field with field strength $(f_\alpha)_{\mu\nu}$, $c_\mu$ is a dynamical $U(1)$ one-form gauge field, and $b_{\mu\nu}$, $\beta_{\mu\nu}$, and $\tilde{\beta}_{\mu\nu}$ are all dynamical $U(1)$ two-form gauge fields. We have also coupled to a background $\mathbb{Z}_N$ two-form gauge field $B_{\mu\nu}$, which probes a $\mathbb{Z}_N$ magnetic one-form symmetry that acts on 't Hooft lines.

Because gauging the discrete one-form symmetry does not change the renormalization group (RG) flow of the gauge coupling $g$, the low energy physics of the theory at $m = 0$ can be deduced from our knowledge of $SU(N)$ adjoint QCD. In particular, for $m = 0$ the theory is IR free for $N_f \geq 7$ flavors, and $N_f = 5$ is believed to be an interacting CFT. Hence, the transition we discuss below will be continuous or first order for the same values of $N_f$ as in the $SU(N)$ theory. However, as we discuss below, the gapped phases for $m \neq 0$ will be rather different from the $SU(N)$ case.

## 4.1 Global symmetries

As with $SU(N)$ adjoint QCD, the zero-form global symmetry of $PSU(N)$ adjoint QCD, Eq. (43), at generic $m$ is $G = SO(N_f) \times \mathbb{Z}_2^T$. The $SO(N_f)$ flavor symmetry acts on the fermions $\psi_J(x)$ in the same way as in Eq. (32). Under time-reversal symmetry, the fermion fields and the $U(N)$ gauge field transform analogously to Eq. (33),

$$
\psi_J(\mathbf{x},t) \to C\gamma^5 \psi_J(\mathbf{x},-t), \qquad \alpha_0(\mathbf{x},t) \to -\alpha_0(\mathbf{x},-t), \qquad \alpha_j(\mathbf{x},t) \to \alpha_j(\mathbf{x},-t). \tag{44}
$$

Again, we have $\mathsf{T}^2 = (-1)^F$. The transformations of $b_{\mu\nu}$ and $c_\mu$ under time-reversal can be inferred from the local constraint, $-\mathrm{Tr}(f_\alpha) = Nb = \mathrm{d}c$. The Lagrange multiplier $\beta_{\mu\nu}$ changes as

$$
\beta_{0j}(\mathbf{x},t) \to \beta_{0j}(\mathbf{x},-t), \qquad \beta_{jk}(\mathbf{x},t) \to -\beta_{jk}(\mathbf{x},-t), \tag{45}
$$

and $\tilde{\beta}_{\mu\nu}$ changes in the same way since $\beta_{\mu\nu} = \tilde{\beta}_{\mu\nu}$ locally. Like in $SU(N)$ adjoint QCD, here magnetic charge is also even under time-reversal while electric charge is odd. Furthermore, $PSU(N)$ adjoint QCD obeys the spin/Kramers relation described in Section 3.1, so all local gauge invariant operators are either bosonic Kramers singlets or fermionic Kramers doublets.

Under parity P, the fields transform as

$$
\begin{aligned}
\psi_J(\mathbf{x},t) \to i\gamma^0 \psi_J(-\mathbf{x},t), \qquad \alpha_0(\mathbf{x},t) \to \alpha_0(-\mathbf{x},t), \qquad \alpha_j(\mathbf{x},t) \to -\alpha_j(-\mathbf{x},t), \\
\beta_{0j}(\mathbf{x},t) \to \beta_{0j}(-\mathbf{x},t), \qquad \beta_{jk}(\mathbf{x},t) \to -\beta_{jk}(-\mathbf{x},t),
\end{aligned}
\tag{46}
$$

so magnetic charge is odd under parity while electric charge is even. Charge conjugation C (for $N > 2$) acts on the fields as

$$
\psi_J(\mathbf{x},t) = -[\psi_J(\mathbf{x},t)]^*, \qquad \alpha_\mu(\mathbf{x},t) \to -[\alpha_\mu(\mathbf{x},t)]^*, \qquad \beta_{\mu\nu}(\mathbf{x},t) \to -\beta_{\mu\nu}(\mathbf{x},t), \tag{47}
$$

where, as in Eq. (35), the complex conjugation acts only on complex numbers and not the Grassmann fields. We see that both electric and magnetic charges are odd under C.

At $m = 0$, the flavor symmetry is enhanced to

$$G_{PSU(N)} = \frac{SU(N_f) \times \mathbb{Z}_{2N_f}}{\mathbb{Z}_{N_f}}. \tag{48}$$

While $SU(N)$ adjoint QCD has a $\mathbb{Z}_{2NN_f}$ axial symmetry, this symmetry is reduced to $\mathbb{Z}_{2N_f}$ in the $PSU(N)$ theory because of the mixed anomaly, Eq. (17), or equivalently, because the theta angle of $PSU(N)$ gauge theory has periodicity $2\pi N$ rather than $2\pi$. However, upon gauging the $\mathbb{Z}_N$ electric one-form symmetry of $SU(N)$ adjoint QCD, the $\mathbb{Z}_N \subset \mathbb{Z}_{2NN_f}$ subgroup of the axial symmetry can be viewed as a non-invertible symmetry in the $PSU(N)$ theory [33, 113, 114].

As long as the subset $G = SO(N_f) \times \mathbb{Z}_2^T$ of the $G_{PSU(N)}$ symmetry at $m = 0$ is preserved, then the only relevant deformation of the $m = 0$ point that respects the symmetry $G$ is the mass in Eq. (43). We are now ready to examine the gapped phases that result from introducing a nonzero fermion mass $m$, which can be deduced from gauging the $\mathbb{Z}_N$ one-form symmetries of the phases in Section 3.2.

## 4.2   $\mathbb{Z}_N$ topological order

For positive $m$, after integrating out the fermions, the gapped phase at low energies is pure $PSU(N)$ gauge theory at $\theta = 0$. As discussed in Section 2.2, the 't Hooft loop $\mathcal{T}(\gamma)$ has a perimeter law in this phase. Because all magnetic charges are deconfined, the full $\mathbb{Z}_N$ magnetic one-form symmetry is spontaneously broken. The low energy physics is described by BF theory at level $N$, Eq. (22).

Since charge $N$ monopoles are screened, this phase may be viewed as a condensate of charge $N$ monopoles, analogous to a superconductor. The analogue of an Abrikosov vortex is an electric flux tube, which is associated with the Wilson surface operator $\mathcal{U}(\Sigma)$ for $b_{\mu\nu}$. A magnetic monopole experiences a dual Aharanov-Bohm effect when it is adiabatically transported around an electric flux tube, as captured by the correlation function,

$$\langle \mathcal{T}(\gamma) \mathcal{U}(\Sigma) \rangle = \exp\left(\frac{2\pi i}{N} \Phi(\gamma, \Sigma)\right), \tag{49}$$

where $\Phi(\gamma, \Sigma)$ is the linking number of the loop $\gamma$ and surface $\Sigma$ in spacetime.

Because $PSU(N)$ adjoint QCD has the zero-form symmetry $G = SO(N_f) \times \mathbb{Z}_2^T$, this $\mathbb{Z}_N$ topological order is also enriched by $G$. Indeed, the $PSU(N)$ monopoles can transform projectively under $G$ since they are not formed by local operators. Thus, there are several possible symmetry fractionalization classes [9, 10] for this phase, which are determined by specifying how the $PSU(N)$ magnetic monopoles transform under $G$. We remark that symmetry fractionalization in adjoint QCD has been studied previously in other contexts [115–118].

Formally, the distinct choices of symmetry fractionalization for the monopoles are given by the pullback of $H_\rho^2(G_b, \Gamma)$ under the projection $G_f \to G_b = G_f/\mathbb{Z}_2^F$ [119, 120] where $G_f$ is the full fermionic symmetry, $\mathbb{Z}_2^F$ is fermion parity, $\Gamma = \mathbb{Z}_N$ is the one-form symmetry, and $\rho$ describes how the symmetry acts on $\Gamma$. To specify $\rho$, we recall that magnetic charge is even under time-reversal while electric charge is odd, and the $SO(N_f)$ symmetry does not permute any line operators. We also note that the full fermionic symmetry $G_f$ should include the appropriate spacetime symmetry. Upon Wick rotating to Euclidean spacetime, we can place the theory on a non-orientable $\text{Pin}^+$ manifold (see Section 3.1), and the Lorentz symmetry acting on the fermions becomes $\text{Pin}^+(4)$. Thus, we have $G_f = \text{Pin}^+(4) \times SO(N_f)$, which corresponds to $G_b = O(4) \times SO(N_f)$, giving

$$H_\rho^2(G_b, \mathbb{Z}_N) = \mathbb{Z}_2 \times \mathbb{Z}_2 \times \mathbb{Z}_2, \tag{50}$$

where we used that $N$ is even. The three generators that can induce changes in the symmetry fractionalization class for $G_b$ are $\pi(w_1)^2$, $\pi w_2$, and $\pi w_2^{SO(N_f)}$. Here, $w_1$ and $w_2$ are the first and second Stiefel-Whitney classes of the spacetime manifold, respectively, and $w_2^{SO(N_f)}$ is the second Stiefel-Whitney class of $SO(N_f)$. In taking the pullback, $\pi(w_1)^2$ and $\pi w_2^{SO(N_f)}$ remain nontrivial, but we take $w_2 = 0$ since this condition is required for a Pin$^+$ manifold. Thus, the symmetry fractionalization classes for the monopoles in this fermionic topological order are labeled by $\mathbb{Z}_2 \times \mathbb{Z}_2$.

We now elucidate the physical meaning of these symmetry fractionalization classes. Under the $SO(N_f)$ symmetry, the monopoles can transform either as tensors or spinors, so the fractionalization class depends on whether the monopoles transform under a $2\pi$ rotation of $SO(N_f)$ by $+1$ or $-1$. If the unit monopole is an $SO(N_f)$ tensor, then taking $B \to B + \pi w_2^{SO(N_f)}$ transforms it into an $SO(N_f)$ spinor. For consistency of the anyon fusion, if the $q_m = 1$ monopole is an $SO(N_f)$ tensor (spinor), then monopoles with odd charge are $SO(N_f)$ tensors (spinors), and monopoles with even charge are always $SO(N_f)$ tensors.

Next, we discuss the possible fractionalization classes of time-reversal symmetry, which may be chosen for the unit monopole independently of the $SO(N_f)$ fractionalization class. If we were studying a (3+1)d time-reversal invariant bosonic SET, we would specify independently whether the point-like anyons are bosons or fermions and whether they transform under time-reversal as Kramers singlets or doublets (i.e., whether $\mathsf{T}^2 = \pm 1$ locally on the anyon). However, we recall from Section 4.1 that adjoint QCD has a spin/Kramers relation—local operators are either bosonic Kramers singlets or fermionic Kramers doublets. Thus, attaching a fermionic Kramers doublet to a point-like anyon should not be viewed as changing the symmetry fractionalization class. This statement is the physical meaning of the restriction $w_2 = 0$ on Pin$^+$ manifolds. Thus, if the unit monopole is a bosonic Kramers singlet or a fermionic Kramers doublet, then we should regard the topological order as having the "trivial" symmetry fractionalization class for time-reversal. The nontrivial class can be induced by the transformation $B \to B + \pi(w_1)^2$, which transmutes a bosonic Kramers singlet (fermionic Kramers doublet) unit monopole into a bosonic Kramers doublet (fermionic Kramers singlet) [121]. Again, for consistency with fusion rules, if the $q_m = 1$ monopole is a bosonic Kramers singlet (doublet), then all monopoles with odd $q_m$ are also bosonic Kramers singlets (doublets) while monopoles with even $q_m$ are always bosonic Kramers singlets (or fermionic Kramers doublets).

Finally, in a general (3+1)d SET, it is possible for the symmetry $G$ to fractionalize on loop excitations [115, 122, 123]. In the $\mathbb{Z}_N$ topological order we study here, the loops excitations are represented by surface operators $\mathcal{U}(\Sigma)$ in spacetime. In this phase, although there is an emergent $\mathbb{Z}_N$ two-form symmetry acting on $\mathcal{U}(\Sigma)$, this symmetry is explicitly broken in $PSU(N)$ gauge theory since the surface operators $\mathcal{U}(\Sigma)$ can be opened and end on a Wilson loop as in Eq. (18). These Wilson loops do not carry any fractional quantum numbers under $G$ and hence are not fractionalized, which simplifies the possible symmetry fractionalization classes considerably. Since the $\mathbb{Z}_N$ electric two-form symmetry has a mixed anomaly with the $\mathbb{Z}_N$ magnetic one-form symmetry, if the loop excitations could transform projectively under $G$, we would have to check whether combinations of symmetry fractionalization for the point-like anyons and loop excitations are anomalous. Because the loops do not transform projectively here, all the possible fractionalization classes discussed above for the monopoles are not anomalous.

## 4.3 $\mathbb{Z}_{N/2}$ topological order

Next, we examine the other gapped phase, which occurs for negative $m$. The nature of this phase can be deduced from Eq. (41) by promoting the background $\mathbb{Z}_N$ two-form gauge field in Eq. (41) to a dynamical field. Thus, the low energy physics of this phase is governed by a

TQFT with effective action,

$$
\begin{aligned}
S_{\text{SET}} = {} & \frac{N(N-1)NN_f/2}{4\pi} \int b \wedge b + \frac{N}{2\pi} \int b \wedge (\mathrm{d}\tilde{a} + B) + \frac{\pi(N^2-1)}{8\pi^2} \int \mathrm{Tr}(F \wedge F) \\
& + \frac{\pi N_f(N^2-1)}{384\pi^2} \int \mathrm{Tr}(R \wedge R),
\end{aligned}
\tag{51}
$$

where $b_{\mu\nu}$ is a dynamical $U(1)$ two-form gauge field, $\tilde{a}_\mu$ is a Lagrange multiplier that constrains $b_{\mu\nu}$ to be a $\mathbb{Z}_N$ gauge field, $B_{\mu\nu}$ is a background $\mathbb{Z}_N$ two-form gauge field, $F$ is the field strength of the background gauge field probing the $SO(N_f)$ flavor symmetry, and $R$ is the curvature two-form. The topological order for the two-form gauge theory TQFT is worked out in Appendix A. The $\mathbb{Z}_N$ magnetic one-form symmetry is spontaneously broken to $\mathbb{Z}_2$, resulting in $\mathbb{Z}_N/\mathbb{Z}_2 \cong \mathbb{Z}_{N/2}$ topological order, and the unbroken $\mathbb{Z}_2$ magnetic one-form symmetry has nontrivial SPT order. For $N = 2$, there is no topological order, so in that case, this phase is a $\mathbb{Z}_2$ one-form SPT state.

Let us understand this topological order from a physical perspective. If $m$ is negative, then integrating out the fermions at low energy results in pure $PSU(N)$ gauge theory with $\theta = \pi N N_f$. Because the theta term is odd under time-reversal symmetry and $\theta$ has periodicity $2\pi N$ in $PSU(N)$ gauge theory, the theory is indeed time-reversal invariant at $\theta = \pi N N_f \sim \pi N$, where we have used that $N_f$ is an odd integer.

For $N$ even, $\theta = \pi N$ is an integer multiple of $2\pi$, and according to Eq. (23), the charges of dyons with a perimeter law are of the form

$$
\left( -\frac{N}{2} q_m, q_m \right),
\tag{52}
$$

for some integer $q_m \bmod N$. As a consistency check, we note that time-reversal symmetry acts on dyon charges as

$$
\mathbb{Z}_2^T: \quad (q_e, q_m) \rightarrow (-q_e, q_m).
\tag{53}
$$

Hence, each dyon with a perimeter law, Eq. (52), is essentially mapped to itself under time-reversal symmetry. A minor caveat is that the $(-N/2, 1)$ dyon is actually mapped to $(N/2, 1)$, but these dyons simply differ by a neutral fermion, so they should be identified with each other since the topological order is fermionic.

The subset of the dyons in Eq. (52) that are associated with genuine loop operators are the purely magnetic charges,

$$
(q_e, q_m) = (0, 2q) \quad \bmod N,
\tag{54}
$$

where $q \in \left\{ 0, 1, \ldots, \frac{N}{2} - 1 \right\}$. The $\mathbb{Z}_N$ magnetic one-form symmetry is therefore spontaneously broken to $\mathbb{Z}_2$ at low energies, resulting in a $\mathbb{Z}_N/\mathbb{Z}_2 = \mathbb{Z}_{N/2}$ topological order.

If we couple to a background field $B_{\mu\nu}$ that probes the unbroken $\mathbb{Z}_2$ one-form symmetry, according to Eq. (27), we obtain the response,

$$
S_{\text{resp}}[B] = \frac{2}{4\pi} \int B \wedge B,
\tag{55}
$$

resulting from the dyon $(-N/2, 1)$ that has a perimeter law but is not a genuine loop operator. Indeed, the symmetry operator for the unbroken $\mathbb{Z}_2$ magnetic one-form symmetry is $\mathcal{U}(\Sigma)^{N/2}$, and if we open this surface, it must end on a loop with electric charge $-N/2$. The response, Eq. (55), indicates that the loop must also have magnetic charge 1.

Now suppose we place the bulk phase on a manifold $X$ with a boundary $\partial X$. The second and third terms of Eq. (51) are the same as in Eq. (41), so the topological order in $X$ is stacked with a fermionic SPT protected by the zero-form symmetry $G = SO(N_f) \times \mathbb{Z}_2^T$. Thus, as in

Section 3.2, a possible boundary state on $\partial X$ that saturates the anomaly for this SPT sector of $G$ is $N_f(N^2 - 1)$ free massless Majorana fermions.

The boundary states associated with the first term of Eq. (51) that preserve the $\mathbb{Z}_N$ magnetic one-form symmetry have been studied previously [79]. The minimal topological order on $\partial X$ must include the deconfined bulk particles (cf. Eq. (54)), a semion, and an anti-semion. The semion can be formed by fusing the anti-semion and a transparent fermion from the bulk. Physically, the anti-semion is the $(-N/2, 1)$ dyon, associated with operator $\mathcal{D}_{(-N/2,1)}(\gamma, \Omega)$ in Eq. (20), which has a perimeter law but is not a genuine loop operator. If the bulk phase, Eq. (51), is placed on a closed manifold, then because the dyon operator $\mathcal{D}_{(-N/2,1)}(\gamma, \Omega)$ is supported on a loop $\gamma$ attached to an open surface $\Omega$, the loop $\gamma$ is always contractible. However, if Eq. (51) is placed on an open manifold $X$ with a boundary $\partial X$, then $\gamma$ can be noncontractible on $\partial X$ with $\Omega$ extending into the bulk. Since $\mathcal{D}_{(-N/2,1)}(\gamma, \Omega)$ has a perimeter law, it will be deconfined on $\partial X$. Fusing the $(-N/2, 1)$ anti-semion with itself generates the dyons in Eq. (54) that contribute to the bulk topological order. Because these dyons braid trivially with all other anyons on $\partial X$, the topological order along $\partial X$ is non-modular and cannot exist in a purely (2+1)d theory, just as in boundary topological orders of Walker-Wang models [124, 125].

Finally, as in Section 4.2, we discuss the possible symmetry fractionalization classes for $G = SO(N_f) \times \mathbb{Z}_2^T$. In the $m < 0$ phase, the $PSU(N)$ monopoles with odd magnetic charge are not part of the bulk topological order. Only the monopoles with even magnetic charge remain (cf. Eq. (54)). The symmetry fractionalization class for the $m < 0$ phase is correlated with the class for the $m > 0$ phase, so the even charge monopoles of the $m < 0$ phase are also not fractionalized under the $G$ symmetry—they are bosonic Kramers singlets (or fermionic Kramers doublets) and $SO(N_f)$ tensors.

Although all the point-like anyons in the topological order of the $m < 0$ phase always transform linearly under the symmetry $G$, the different choices of symmetry fractionalization for the unit monopole will result in different SPT responses in the $m < 0$ phase.[14] If the unit monopole is a bosonic Kramers singlet and $SO(N_f)$ tensor in the $m > 0$ phase, then in the $m < 0$ phase, the minimal boundary topological order described above is enough to match anomaly inflow from the bulk. If we take $B \to B + \pi(w_1)^2$ so that the unit monopole is a bosonic Kramers doublet, then the bulk response in Eq. (55) will be modified, thus changing the nature of the SPT order. To be specific, fermionic SPTs with time-reversal symmetry such that $\mathsf{T}^2 = (-1)^F$ are classified by a topological invariant $\nu \in \mathbb{Z}_{16}$ [85, 126–128], and taking $B \to B + \pi(w_1)^2$ changes this topological invariant by $\Delta \nu = \pm 4$ [28, 118]. A boundary topological order consistent with this modified bulk response is $\{1, s_1\} \otimes \{1, s_2\} \otimes \{1, f\}$, where $s_1$ and $s_2$ are semions transforming under time-reversal as $\mathsf{T}^2 = \pm i$ and $f$ is fermionic Kramers doublet [126, 128]. Similarly, if the unit monopole of the $m > 0$ phase is an $SO(N_f)$ spinor, then the boundary topological order must contain a semion that transforms as a spinor under the $SO(N_f)$ symmetry [129, 130].

To summarize, the negative $m$ phase has $\mathbb{Z}_{N/2}$ topological order that results from spontaneously breaking the $\mathbb{Z}_N$ magnetic one-form symmetry to $\mathbb{Z}_2$. This topological order is enriched by the $G = SO(N_f) \times \mathbb{Z}_2^T$ zero-form symmetry and the unbroken $\mathbb{Z}_2$ magnetic one-form symmetry. The topological order is stacked with a nontrivial SPT for the symmetry $G$ and the $\mathbb{Z}_2$ one-form symmetry, and the precise nature of this SPT is determined by the choice of symmetry fractionalization class for the unit monopole. The transition between this phase and the positive $m$ phase discussed in Section 4.2 is depicted schematically in Figure 2. While we have focused on $SU(N)/\mathbb{Z}_N$ gauge theory coupled to $N_f$ Majorana fermions in the adjoint representation, similar constructions exist for other gauge groups, which we discuss in Appendix E.

---

[14]Similar observations were described in Ref. [28].

$$G \times \mathbb{Z}_2^{(1)} \text{ SPT}$$

$$\mathbb{Z}_{N/2} \text{ TO} \qquad\qquad \mathbb{Z}_N \text{ TO}$$

$$\underline{\phantom{xxxxxxxxxxx} \bullet \phantom{xxxxxxxxxxx}}$$

$$m < 0 \qquad\qquad m > 0$$

Figure 2: Schematic depiction of the phase diagram of $PSU(N) = SU(N)/\mathbb{Z}_N$ adjoint QCD, Eq. (43), for $N$ even and $N_f$ odd flavors of Majorana fermions as a function of the mass $m$. The $m > 0$ phase spontaneously breaks the full $\mathbb{Z}_N^{(1)}$ magnetic one-form symmetry at low energies, giving rise to $\mathbb{Z}_N$ topological order described by BF theory at level $N$, Eq. (22). For $m < 0$, the low energy physics is described by Eq. (51). The $\mathbb{Z}_N^{(1)}$ magnetic one-form symmetry, is spontaneously broken to $\mathbb{Z}_2^{(1)}$, resulting in $\mathbb{Z}_{N/2}$ topological order. Additionally, the $m < 0$ phase is stacked with a nontrivial fermionic SPT for the unbroken $\mathbb{Z}_2^{(1)}$ and $G = SO(N_f) \times \mathbb{Z}_2^T$ symmetries. The $G$ symmetry may be fractionalized on the point-like anyon excitations of the bulk topological order in the $m > 0$ phase. These different choices of symmetry fractionalization in the $m > 0$ phase are correlated with the particular $G \times \mathbb{Z}_2^{(1)}$ SPT order realized in the $m < 0$ phase.

### 4.4 String tension critical exponent

Since we have an example of a critical point between phases with different patterns of one-form symmetry breaking, it is natural to study critical exponents for loop operators. In the $PSU(N)$ theory coupled to adjoint fermions, the basic 't Hooft line $\mathcal{T}(\gamma)$ has an area law in the negative mass phase, but this monopole becomes deconfined both at the critical point, $m = 0$, and in the positive mass phase. As $m$ approaches zero from the negative mass phase, the string tension of $\mathcal{T}(\gamma)$ is expected to vanish as

$$\sigma \sim |m|^\mu, \tag{56}$$

for some critical exponent $\mu$, which can be reliably computed for $N_f > 5$ so that the $m = 0$ theory is infrared free [131,132].

For $m < 0$, the low energy physics, well below the fermion mass $|m|$, is governed by pure $PSU(N)$ Yang-Mills at $\theta = \pi N N_f$. On grounds of dimensional analysis, the string tension of $\mathcal{T}(\gamma)$ scales as

$$\sigma \sim (\Lambda_{\text{YM}})^2, \tag{57}$$

where $\Lambda_{\text{YM}}$ is the energy scale dynamically generated in the pure non-Abelian gauge theory. To compute the critical exponent $\mu$, we must then relate $\Lambda_{\text{YM}}$ to the mass scale $|m|$ of the fermions. For energies far above $|m|$, the beta function for $g$ is well-approximated by Eq. (42). In contrast, at energies far below $|m|$, the fermions may be integrated out, and the beta function is governed by that of pure Yang-Mills. To estimate the string tension, we approximate the beta function as

$$\beta(g) = \frac{dg}{d\ell} = \begin{cases} (c_{\text{YM}} - c_{\text{M}}) g^3, & \Lambda > |m|, \\ c_{\text{YM}} g^3, & \Lambda < |m|, \end{cases} \tag{58}$$

where $\Lambda = \Lambda_{\text{UV}} e^{-\ell}$ is the energy associated with scale $\ell$ and $\Lambda_{\text{UV}}$ is a high energy reference scale.

Integrating Eq. (58) from $\Lambda_{\text{UV}}$ to the mass scale gives

$$g^2(|m|) = \frac{g^2(\Lambda_{\text{UV}})}{1 + 2(c_{\text{M}} - c_{\text{YM}}) g^2(\Lambda_{\text{UV}}) \ln(\Lambda_{\text{UV}}/|m|)}. \tag{59}$$

We also integrate Eq. (58) from $|m|$ to an energy scale $\Lambda_0$ below $|m|$, resulting in

$$g^2(\Lambda_0) = \frac{g^2(|m|)}{1 - 2\,c_{\mathrm{YM}}\,g^2(|m|)\ln(|m|/\Lambda_0)} \, . \tag{60}$$

The scale $\Lambda_{\mathrm{YM}}$ is determined by setting $1/g^2(\Lambda_{\mathrm{YM}}) = 0$ using Eq. (60). Solving for $\Lambda_{\mathrm{YM}}$ and using the expression in Eq. (59) for $g^2(|m|)$ gives

$$\Lambda_{\mathrm{YM}} = |m|\exp\left(-\frac{1}{2\,c_{\mathrm{YM}}\,g^2(|m|)}\right) = \Lambda_{\mathrm{UV}}\,e^{-1/(2\,c_{\mathrm{YM}}\,g^2(\Lambda_{\mathrm{UV}}))}\left(\frac{|m|}{\Lambda_{\mathrm{UV}}}\right)^{c_{\mathrm{M}}/c_{\mathrm{YM}}} \, . \tag{61}$$

Although this calculation is based on Eq. (58), this approximation becomes reliable in the limit $|m|/\Lambda_{\mathrm{UV}} \to 0$ if the $m = 0$ point is infrared free since the coupling $g$ remains small at energy scales near $|m|$. Indeed, Refs. [131, 132] numerically solved the coupled RG equations for $g$ and $m$ to obtain $\Lambda_{\mathrm{YM}}/\Lambda_{\mathrm{UV}}$ as a function of $|m|/\Lambda_{\mathrm{UV}}$, which agrees with Eq. (61) for small $|m|/\Lambda_{\mathrm{UV}}$.

Thus, the critical exponent $\mu$ for the string tension of the 't Hooft loop $\mathcal{T}(\gamma)$ is

$$\mu = \frac{2\,c_{\mathrm{M}}}{c_{\mathrm{YM}}} = \frac{4}{11}\sum_{\mathcal{R}}\frac{I_{\mathcal{R}}}{I_{\mathrm{adj}}}\,N_f(\mathcal{R}) = \frac{4N_f}{11} \, , \tag{62}$$

where we specialized to $N_f$ flavors of adjoint fermions in the last step. Note that we did not have to specify a particular gauge group, so this critical exponent is the same for the adjoint QCD transitions with other gauge groups discussed in Appendix E. It is interesting to compare this exponent with the prediction $\mu = 1/2$ from mean string field theory [27]. For the values of $N_f$ where we expect Eq. (62) to be valid (i.e., $N_f \geq 7$), we have $\mu \geq 28/11 \approx 2.55$. Thus, in these cases, the string tension vanishes more strongly as the critical point is approached than mean string field theory predicts.

# 5 SET to non-invertible SSB transition

As we have established, massless $PSU(N)$ adjoint QCD with a sufficiently large odd number of flavors $N_f$ can appear at a critical point between two SETs with different topological orders, provided that both the $SO(N_f)$ flavor symmetry and time-reversal symmetry are preserved. Recent work has introduced non-invertible analogues of time-reversal symmetry [33, 34], which are also symmetries of massless $PSU(N)$ adjoint QCD. This observation then raises the question of what kinds of phases can emerge if we consider deformations of the massless point that respect the non-invertible time-reversal symmetry and $SO(N_f)$ flavor symmetry.

In this section, we will analyze the theory with action,

$$S_n = S_0 + \frac{\pi n}{8\pi^2}\int\left\{\mathrm{Tr}[(f_\alpha + b\,\mathbb{I}_N)\wedge(f_\alpha + b\,\mathbb{I}_N)] - [\mathrm{Tr}(f_\alpha) + Nb]\wedge[\mathrm{Tr}(f_\alpha) + Nb]\right\}, \tag{63}$$

where $S_0$ is the $PSU(N)$ adjoint QCD action defined in Eq. (43). The additional term is a $PSU(N)$ theta term with $\theta = \pi n$. While $S_0$ is invariant under the standard invertible time-reversal transformation, the additional theta term in not unless $n \in N\mathbb{Z}$.

Below, we will review the notion of non-invertible time-reversal symmetry and show that Eq. (63) respects this symmetry. Imposing this non-invertible time-reversal symmetry and the $SO(N_f)$ flavor symmetry, for large enough odd $N_f$, we can again have a direct continuous transition tuned by a single parameter. For $N$ and $N_f$ both odd, as we demonstrate below, this transition can be between a topologically ordered phase, which breaks the magnetic one-form

symmetry to a subgroup, and a phase that breaks the non-invertible time-reversal symmetry spontaneously. To keep the discussion simple, we will primarily focus on the case $N = k^2$ and $n = 2k$, where $k > 1$ is odd, though these choices for $N$ and $n$ are not the only ones that give transitions between a topologically ordered phase and a non-invertible SSB phase.

## 5.1 Non-invertible time-reversal

To establish the non-invertible symmetry, it is useful to define the following transformations on a theory with a $\mathbb{Z}_N$ one-form global symmetry with action $S[B]$,

$$
\begin{aligned}
\mathbf{S}: \quad & S[B] \to S[b] + \frac{N}{2\pi} \int b \wedge B, \\
\mathbf{T}: \quad & S[B] \to S[B] + \frac{N}{4\pi} \int B \wedge B, \\
\mathbf{C}: \quad & S[B] \to S[-B],
\end{aligned}
\tag{64}
$$

where $B_{\mu\nu}$ is a background $\mathbb{Z}_N$ two-form gauge field for the $\mathbb{Z}_N$ one-form global symmetry and $b_{\mu\nu}$ is a dynamical $\mathbb{Z}_N$ two-form gauge field. These transformations obey

$$
\mathbf{S}^2 = \mathbf{C}, \qquad \mathbf{C}^2 = 1, \qquad \mathbf{T}^N = 1.
\tag{65}
$$

To summarize, $\mathbf{S}$ denotes gauging of the $\mathbb{Z}_N$ one-form symmetry, $\mathbf{T}$ stacks a $\mathbb{Z}_N$ one-form SPT, and $\mathbf{C}$ simply changes the sign of the background field.

A non-invertible time-reversal transformation can be defined by [33, 34]

$$
\mathbf{K}_n = \mathbf{C}\mathbf{S}\mathbf{T}^{(N-1)n}\mathbf{S}\mathbf{K},
\tag{66}
$$

where $\mathbf{K}$ is the standard invertible time-reversal transformation and $n$ is an integer mod $N$. From the relations, Eq. (65), we observe that if $n \in N\mathbb{Z}$, then $\mathbf{K}_n$ reduces to $\mathbf{K}$. We recall that the theta term is odd under $\mathbf{K}$, and we observe that $\mathbf{C}\mathbf{S}\mathbf{T}^{(N-1)n}\mathbf{S}$ shifts $\theta \to \theta + 2\pi n$. Hence, the non-invertible time-reversal transformation maps

$$
\mathbf{K}_n: \quad \theta \to -\theta + 2\pi n.
\tag{67}
$$

Since $\theta$ has periodicity $2\pi N$, the values of $\theta$ that are invariant under $\mathbf{K}_n$ are $\theta = \pi n$ and $\theta = \pi(n + N)$. Because the theta term added to Eq. (63) is such that $\theta = \pi n$, $\mathbf{K}_n$ indeed leaves Eq. (63) invariant as promised.

For $n \notin N\mathbb{Z}$, while the transformation $\mathbf{K}_n$ leaves Eq. (63) invariant, it cannot be associated with a unitary operator because it involves $\mathbf{S}$. However, $\mathbf{K}_n$ may be associated with a non-invertible defect $\mathsf{T}_n = \mathsf{D}_n \mathsf{T}$ which is topological. Here, $\mathsf{T}$ is an interface that reverses orientation,[15] and $\mathsf{D}_n$ is defined as [33, 34, 113, 114]

$$
\mathsf{D}_n = \int \mathcal{D}a_1 \mathcal{D}a_2 \exp\left[ i \oint_Y \left( \frac{N(N-1)n}{4\pi} a_1 \wedge da_1 + \frac{N}{2\pi} a_1 \wedge da_2 + \frac{N}{2\pi} a_2 \wedge b \right) \right],
\tag{68}
$$

where $Y$ is a closed three-dimensional manifold in spacetime while $(a_1)_\mu$ and $(a_2)_\mu$ are dynamical $U(1)$ one-form gauge fields defined only on $Y$. Hence, $\mathsf{T}_n$ is constructed by decorating $\mathsf{T}$ with a particular fractional quantum Hall state. The defect $\mathsf{T}_n$ obeys the fusion rules,

$$
\begin{aligned}
\mathsf{T}_n \times (\mathsf{T}_n)^\dagger &= \mathsf{D}_n \times \mathsf{D}_{-n} = (\mathcal{Z}_N)_{N(N-1)n} \mathcal{C}_0, \\
\mathsf{T}_n \times \mathsf{T}_n &= \mathsf{D}_n \times \mathsf{D}_{-n} \times \mathsf{T}^2 = (\mathcal{Z}_N)_{N(N-1)n} \mathcal{C}_0 (-1)^F,
\end{aligned}
\tag{69}
$$

---

[15]At $\theta = 0$, $\mathsf{T}$ is an invertible symmetry, and the operator associated with $\mathsf{T}$ transforms the fields as described in Section 4.1.

where $(-1)^F$ is fermion parity, $(\mathcal{Z}_N)_{N(N-1)n}$ is the partition function for $\mathbb{Z}_N$ Chern-Simons theory at level $N(N-1)n$,

$$(\mathcal{Z}_N)_{(N-1)nN} = \int \mathcal{D}a_1 \mathcal{D}a_2 \exp\left[ i \oint_Y \left( \frac{N}{2\pi} a_1 \wedge \mathrm{d}a_2 + \frac{N(N-1)n}{4\pi} a_1 \wedge \mathrm{d}a_1 \right) \right], \qquad (70)$$

and $\mathcal{C}_0$ is a condensation defect [133–137],

$$\mathcal{C}_0 = \int \mathcal{D}a_1 \mathcal{D}a_2 \exp\left( i \oint_Y \frac{N}{2\pi}(a_1 \wedge \mathrm{d}a_2 + a_1 \wedge b) \right), \qquad (71)$$

where the $\mathbb{Z}_N$ one-form symmetry is gauged only along $Y$.

The defect $\mathsf{T}_n$ is topological in the theory, Eq. (63), for any value of the fermion mass $m$, so we refer to $\mathsf{T}_n$ as a non-invertible time-reversal symmetry. In the remainder of Section 5, we will examine the phases of Eq. (63) as a function of the $\mathsf{T}_n$ preserving mass $m$. Importantly, the non-invertible symmetry can have important non-perturbative constraints on the phase diagram, a notion explored for (3+1)d systems in a number of recent works [64, 138–144]. As we demonstrate in Appendix F, for $N$ odd and $n$ such that $\gcd(N, n) > 1$, the non-invertible time-reversal symmetry and $\mathbb{Z}_N$ magnetic one-form symmetry have a mixed anomaly in the sense that no single ground state can preserve both these symmetries. The phase diagram we find, Figure 3, is consistent with this constraint.

## 5.2  SET phase

For $m > 0$, at energies far below the mass scale of the fermions, the physics is governed by pure $PSU(N)$ gauge theory with $\theta = \pi n$. Taking $n$ even with $n = 2k$, the low energy physics, as established in Section 2.2, is described by the TQFT,

$$S_{\mathrm{TQFT}}[(N-1)k, \tilde{a}, b, B] = \frac{N(N-1)k}{4\pi} \int b \wedge b + \frac{N}{2\pi} \int b \wedge (\mathrm{d}\tilde{a} + B), \qquad (72)$$

where $\tilde{a}_\mu$ is a dynamical $U(1)$ one-form gauge field, $b_{\mu\nu}$ is a dynamical $U(1)$ two-form gauge field, and $B_{\mu\nu}$ is a background $\mathbb{Z}_N$ two-form gauge field. As discussed in Section 2.2 and Appendix A, the physics of this phase depends on $\gcd(N, k)$. If $\gcd(N, k) = 1$, then this phase is an SPT protected by the $\mathbb{Z}_N$ magnetic one-form symmetry and is also invariant under the non-invertible time-reversal symmetry. Otherwise, there is topological order. For $n = 0$, the time-reversal symmetry is invertible, and this phase has $\mathbb{Z}_N$ topological order described by BF theory with level $N$.

To focus on a class of examples in which this phase has topological order enriched by a *non-invertible* time-reversal symmetry, we take $N = k^2$ with $k > 1$ odd, which we assume for the remainder of Section 5.2. Within the $m > 0$ phase, the $\mathbb{Z}_N = \mathbb{Z}_{k^2}$ magnetic one-form symmetry is spontaneously broken to $\mathbb{Z}_k$ at low energies, resulting in $\mathbb{Z}_{k^2}/\mathbb{Z}_k \cong \mathbb{Z}_k$ topological order, and the unbroken $\mathbb{Z}_k$ one-form symmetry has nontrivial SPT order. This topological order is also enriched by the unbroken $SO(N_f)$ symmetry.

To understand the $\mathbb{Z}_k$ topological order more physically, we observe from Eq. (23) that the dyons in this phase with a perimeter law have charges of the form $(-k q_m, q_m)$ for an integer $q_m \bmod N$. The non-invertible time-reversal symmetry $\mathsf{T}_{2k}$ maps the charges of a dyon as

$$(q_e, q_m) \rightarrow (-q_e - 2k q_m, q_m). \qquad (73)$$

Thus, each dyon with a perimeter law in this phase is mapped to itself (up to a neutral fermion) under $\mathsf{T}_{2k}$.

The dyons with a perimeter law that are also associated with genuine loop operators are the purely magnetic charges,

$$(q_e, q_m) = (0, k\, q) \mod k^2\,, \tag{74}$$

where $q \in \{0, 1, \ldots, k-1\}$. Thus, there are $k$ anyon particles associated with the 't Hooft loops $\mathcal{T}(\gamma)^{kq}$. These 't Hooft loops can also have nontrivial correlation functions with the electric flux tubes associated with the surface operators $\mathcal{U}(\Sigma)$,

$$\left\langle \mathcal{T}(\gamma)^k \mathcal{U}(\Sigma)\right\rangle = \exp\left(\frac{2\pi i}{k}\, \Phi(\gamma, \Sigma)\right)\,, \tag{75}$$

where $\Phi(\gamma, \Sigma)$ is the linking number of the loop $\gamma$ and surface $\Sigma$ in spacetime.

Following the analysis of the TQFT, Eq. (72), in Appendix A, the response to a background field $B_{\mu\nu}$ for the unbroken $\mathbb{Z}_k \subset \mathbb{Z}_{k^2}$ one-form symmetry is

$$S_{\text{resp}}[B] = \frac{k}{4\pi}\int B \wedge B\,. \tag{76}$$

To interpret this response, we note that the symmetry operator for the unbroken $\mathbb{Z}_k$ one-form symmetry is $\mathcal{U}(\Sigma)^k$, which describes the worldsheet of an electric flux loop with flux $-k$. When this operator ends on a loop, the loop must carry electric charge $-k$, and the response indicates that it also carries magnetic charge 1. Indeed, this operator will be nontrivial in the low energy limit only if the loop on which it ends has a perimeter law. Hence, the physical meaning of Eq. (76) is that the $(-k, 1)$ dyon, which is not genuine loop operator, has a perimeter law.

## 5.3 Non-invertible SSB

Next, we determine the phase that arises for $m < 0$. In this case, at energies well below the scale of the fermion mass $|m|$, the fermions may be integrated out to give an effective action of

$$S_{\text{eff}} = S_{PSU(N)}[\pi(n + N N_f), B] + \frac{\pi(N^2-1)}{8\pi^2}\int \text{Tr}(F \wedge F) + \frac{\pi N_f(N^2-1)}{384\pi^2}\int \text{Tr}(R \wedge R)\,, \tag{77}$$

where $F_{\mu\nu}$ is the field strength of the background field $A_\mu$ for the $SO(N_f)$ flavor symmetry, $R$ is the curvature two-form (see Appendix C), and $S_{PSU(N)}[\pi(n + N N_f), B]$ is the action for pure $PSU(N)$ gauge theory (cf. Eqs. (15) and (17)) with $\theta = \pi(n + N N_f)$, which is equivalent to $\theta = \pi(n + N)$ for odd $N_f$. For odd $N$ and even $n$, then $\theta$ is an odd multiple of $\pi$. As we discuss below, this phase spontaneously breaks the non-invertible time-reversal symmetry $\mathsf{T}_n$. We can then ignore the theta terms for the background fields $F_{\mu\nu}$ and $R$ in Eq. (77). Since this phase does not have unbroken time-reversal symmetry (invertible or not), the theta terms for these background fields may be continuously tuned to zero without encountering a phase transition.

As we reviewed in Section 2.2, there is evidence that at $\theta = \pi\tilde{n}$ for odd integer $\tilde{n}$, the ground states at $\theta = \pi(\tilde{n} - 1)$ and $\theta = \pi(\tilde{n} + 1)$ are degenerate, which may be established more rigorously for large $N$ [75]. This IR behavior may be interpreted as the spontaneous breaking of the non-invertible time-reversal symmetry associated with the operator $\mathsf{T}_n$, which exchanges the ground state(s) at $\theta = \pi(\tilde{n} - 1)$ with the state(s) at $\theta = \pi(\tilde{n} + 1)$ for $\tilde{n} = n + N$. In the special case $n = 0$ (or equivalently, $\tilde{n} = N$), the time-reversal symmetry is invertible and spontaneously broken in this phase, at least for large enough $N$.

For the remainder of Section 5.3, let us take $N = k^2$ and $n = 2k$ for odd $k > 1$. The time-reversal symmetry operator $\mathsf{T}_{2k}$ is non-invertible, and we can still in principle access the

$$
\begin{array}{ccc}
\text{Non-invertible} & & \mathbb{Z}_k^{(1)}\ \text{SPT} \\
\text{SSB} & \bullet & \mathbb{Z}_k\ \text{TO} \\
\hline
m < 0 & & m > 0
\end{array}
$$

Figure 3: Schematic phase diagram of $PSU(N) = SU(N)/\mathbb{Z}_N$ adjoint QCD with a theta term $\theta = \pi n$, Eq. (63), for $N_f$ odd flavors of Majorana fermions as a function of the fermion mass $m$. Here, we take $N = k^2$ and $n = 2k$ with $k > 1$ odd. In the $m > 0$ phase, described by the TQFT in Eq. (72), the $\mathbb{Z}_N = \mathbb{Z}_{k^2}$ magnetic one-form symmetry is spontaneously broken to $\mathbb{Z}_k$, resulting in $\mathbb{Z}_k$ topological order. The unbroken $\mathbb{Z}_k$ one-form symmetry also has nontrivial SPT order. For $m < 0$, the $\mathbb{Z}_N$ one-form symmetry is unbroken, but the non-invertible time-reversal symmetry associated with operator $\mathsf{T}_{2k} = \mathsf{D}_{2k}\mathsf{T}$ is spontaneously broken, resulting in two ground states that are distinct $\mathbb{Z}_N$ one-form SPTs with responses given by Eq. (81). The domain walls in this phase obey the non-invertible fusion rules in Eq. (69). The $SO(N_f)$ flavor symmetry remains unbroken throughout the phase diagram.

large $N$ limit by taking $k$ large. We then have $\tilde{n} = 2k + N = 2k + k^2$. The TQFTs describing the states at $\theta = \pi(\tilde{n} \pm 1)$ are

$$
S^{\pm} = \frac{N(N-1)(2k+N\pm 1)}{8\pi}\int b \wedge b + \frac{N}{2\pi}\int b \wedge (\mathrm{d}\tilde{a} + B), \tag{78}
$$

where $\tilde{a}_\mu$ is a dynamical $U(1)$ one-form gauge field, $b_{\mu\nu}$ is a dynamical $U(1)$ two-form gauge field, and $B_{\mu\nu}$ is a background $\mathbb{Z}_N$ gauge field. The ground states described by these TQFTs become degenerate at $\theta = \pi\tilde{n} = \pi(2k + N)$, and they are exchanged by the non-invertible time-reversal symmetry $\mathsf{T}_{2k}$. Indeed, the dyons with a perimeter law at $\theta = \pi(2k + N \pm 1)$ are

$$
(q_e, q_m) = \left(-\frac{N + 2k \pm 1}{2} q_m, q_m\right) \quad \mathrm{mod}\ k^2. \tag{79}
$$

According to Eq. (73), the non-invertible time-reversal symmetry $\mathsf{T}_{2k}$ maps these dyons as

$$
\left(-\frac{N + 2k \pm 1}{2} q_m, q_m\right) \rightarrow \left(-\frac{N + 2k \mp 1}{2} q_m, q_m\right). \tag{80}
$$

Thus, at $\theta = \pi(2k+N)$ the non-invertible time-reversal symmetry $\mathsf{T}_{2k}$ is spontaneously broken. The domain walls obey the non-invertible fusion rules of Eq. (69).

There is no topological order for either of the TQFTs[16] in Eq. (78) because of the identity[17] $\gcd(N, (\tilde{n} \pm 1)/2) = \gcd(k^2, (k^2 + 2k \pm 1)/2) = 1$, which holds for any integer $k$. Thus, the $\mathbb{Z}_N$ magnetic one-form symmetry is unbroken, and the ground state degeneracy in this phase is two on any manifold. Integrating out the dynamical fields in Eq. (78), we obtain responses for the background $\mathbb{Z}_{k^2}$ two-form gauge field $B_{\mu\nu}$ of

$$
S^{\pm}_{\mathrm{SPT}}[B] = -\frac{k^2\,2\,(2k \mp 1)}{4\pi}\int B \wedge B, \tag{81}
$$

so the degenerate ground states are $\mathbb{Z}_{k^2}$ one-form SPTs.

---

[16]This statement does not always hold for generic odd $N$ and $k$. It is also possible for one of the TQFTs to have topological order so that the $\mathbb{Z}_N$ magnetic one-form symmetry is spontaneously broken to a subgroup. But since $k + (N-1)/2$ and $k + (N+1)/2$ are coprime, if $S_{\pm}$ has topological order, then $S_{\mp}$ necessarily does not.

[17]This identity follows from

$$
1 = k^2(2k + 4 \mp 1) + \left(\frac{k^2 + 2k \pm 1}{2}\right)(-2)(2k \mp 1).
$$

To summarize the conclusions of Section 5, we have found that for $N$ odd, sufficiently large odd $N_f$, and even $n$, there can be a continuous transition between a topologically ordered phase and a phase that spontaneously breaks a non-invertible time-reversal symmetry. If we take $N = k^2$ and $n = 2k$ with $k > 1$ odd, the $m < 0$ phase spontaneously breaks the non-invertible time-reversal symmetry $\mathsf{T}_{2k}$ while the $\mathbb{Z}_{k^2}$ one-form symmetry is unbroken. The $m > 0$ phase preserves the non-invertible time-reversal symmetry but spontaneously breaks the magnetic $\mathbb{Z}_{k^2}$ one-form symmetry to $\mathbb{Z}_k$, resulting in $\mathbb{Z}_k$ topological order, and the unbroken $\mathbb{Z}_k$ one-form symmetry also has nontrivial SPT order. The $SO(N_f)$ flavor symmetry is unbroken in both phases. For large enough $N_f$, there is a continuous transition between these phases at $m = 0$ (see Figure 3). In the special case $n = 0$, the time-reversal symmetry is invertible, and the $m > 0$ phase has $\mathbb{Z}_N$ topological order while the $m < 0$ phase spontaneously breaks the invertible time-reversal symmetry.

## 5.4 Critical exponents

Following the methods of Section 4.4 and Refs. [29, 131, 132], the critical exponents for this transition may be easily computed if $N_f$ is large enough that the $m = 0$ point is IR free. For the $m > 0$ phase, the 't Hooft loop has an area law, and following the same reasoning as in Section 4.4, we find that the string tension $\sigma$ for small $m$ behaves as $\sigma \sim |m|^\mu$ with $\mu$ given in Eq. (62).

Turning our attention to critical exponents associated with the non-invertible time-reversal symmetry, we note that an operator odd under $\mathsf{T}_n$, which can serve as a local order parameter, is

$$\phi(x) = \sum_{J=1}^{N_f} i \operatorname{Tr}\left[\psi_J^T(x) \mathcal{C}\gamma^5 \psi_J(x)\right]. \tag{82}$$

Unlike the $\mathbb{Z}_N$ magnetic one-form global symmetry, the non-invertible zero-form symmetry has a local order parameter, and thus, the kinds of critical exponents we can study for this symmetry have closer analogues in conventional Landau theory. Because the critical point is IR free, the order parameter has scaling dimension $\Delta_\phi = 3$, so its two-point function scales as [29]

$$\langle \phi(x)\phi(x')\rangle \sim \frac{1}{|x-x'|^6}, \tag{83}$$

at the critical point.

Next, we examine the critical exponent for how the order parameter vanishes as the critical point is approached from the ordered phase. For $m < 0$, consider weakly perturbing the action $S_n$, Eq. (63), by a term that explicitly breaks the non-invertible time-reversal symmetry,

$$S_n \to S_n + \int d^4x \, h_\epsilon \, \phi(x), \tag{84}$$

where the coefficient $h_\epsilon$ is small, $|h_\epsilon| \ll |m|$. Here, $h_\epsilon$ is the analogue of an external symmetry-breaking field. At low energies, where the fermions may be integrated out, the effect of the perturbation in Eq. (84) is to modify the theta angle of the $m < 0$ phase to

$$\theta = \pi(n+N) \to \pi(n+N) + \frac{h_\epsilon}{|m|} N N_f. \tag{85}$$

Hence, for small $|m|$, we expect the expectation value of the order parameter to scale as [29]

$$\langle \phi(x)\rangle \sim \frac{1}{|m|}\left\langle \operatorname{Tr}\left(\varepsilon^{\mu\nu\lambda\sigma} f_{\mu\nu} f_{\lambda\sigma}\right)\right\rangle \sim \frac{(\Lambda_{\text{YM}})^4}{|m|} \sim |m|^{(8N_f/11)-1}, \tag{86}$$

where we used Eq. (61). This critical exponent is analogous to $\beta = (8N_f/11) - 1$ in Landau theory. For similar reasons, if we add the symmetry-breaking perturbation $h_\epsilon$ but with $m = 0$, we obtain

$$\langle \phi(x) \rangle \sim |h_\epsilon|^{(8N_f/11)-1}, \tag{87}$$

which is analogous to an exponent of $\delta = 11/(8N_f - 11)$ in Landau theory. These critical exponents characterize the universality class of the transition and highlight its analogy with conventional Landau transitions despite that both the phases and the transition are beyond Landau.

# 6  Discussion

In this work, we have introduced two families of exotic transitions between phases that break different generalized symmetries, both of which involve $PSU(N) = SU(N)/\mathbb{Z}_N$ gauge theory coupled to $N_f$ odd flavors of Majorana fermions in the adjoint representation. Our first example is a critical point between SET phases that have distinct topological orders—a phase with $\mathbb{Z}_N$ topological order and another with $\mathbb{Z}_{N/2}$ topological order for $N$ even. While we have focused on this transition for a $PSU(N)$ gauge group, a topological transition of this type can occur for adjoint QCD with other gauge groups, as discussed in Appendix E. To our knowledge, this theory provides the first clear example in (3+1)d of a model with an exact one-form symmetry that displays a continuous transition between phases with different patterns of one-form symmetry breaking.

The second kind of unconventional transition we have explored is a continuous transition between a topologically ordered phase, which spontaneously breaks a discrete one-form symmetry, and a phase that spontaneously breaks a non-invertible time-reversal symmetry, providing an analogue of deconfined quantum criticality for generalized symmetries. This critical point represents a "beyond Landau" transition between phases that also lie beyond Landau. Taken together, our two examples of exotic transitions can serve as guides for discovering other topological critical points and developing a more general theory classifying these transitions.

There are several promising directions for future work. One is to analyze symmetry fractionalization of the non-invertible time-reversal symmetry in the SET phase discussed in Section 4.2. While fractionalization of non-invertible symmetries has been studied in some examples in (2+1)d [145], there is currently no general framework for fractionalization of non-invertible symmetries. The (3+1)d $\mathbb{Z}_k$ topological order enriched by non-invertible time-reversal symmetry in Section 5.2 provides a concrete example where this analysis can be done.

Another possible avenue is to investigate the fate of the SET phase discussed in Section 4.3 when the magnetic one-form symmetry is explicitly broken. Concretely, this question can be explored by studying the lattice model of Appendix D, where the degree to which the one-form symmetry is explicitly broken is determined by the two-form gauge coupling $\tilde{g}^2$. For small enough $\tilde{g}^2$, the phase discussed in Section 4.3 will have topological orders both in the bulk and on the boundary. At large $\tilde{g}^2$, these topological orders are expected to disappear. A natural question is whether the transitions must occur simultaneously, or if there can be an analogue of the extraordinary transition for one-form symmetries. Similarly, it would also be interesting to study boundary criticality in the transition between the $\mathbb{Z}_{N/2}$ and $\mathbb{Z}_N$ topologically ordered phases.

Finally, all the transitions discussed in this work involve magnetic one-form symmetries. It would be interesting to develop a dual description in which these magnetic one-form symmetries are mapped to electric one-form symmetries. Such a duality would not only offer a complementary perspective on the dynamics of these transitions, but would also constitute a rare example of a non-supersymmetric duality in (3+1)d.

**Note:** While completing this manuscript, we became aware of a related work [146] that also studies transitions of invertible one-form symmetries in theories with local quantum fields but without imposing a flavor symmetry on matter fields.

## Acknowledgments

The author is especially grateful to Eduardo Fradkin for numerous insightful discussions and for comments on the manuscript. We also thank Fiona Burnell, Junyi Cao, Clay Córdova, Hart Goldman, Yin-Chen He, and Ramanjit Sohal for useful discussions, as well as John McGreevy and Matthew C. O'Brien for feedback on the draft.

**Funding information** This work was supported in part by the National Science Foundation under grant DMR-2225920 at the University of Illinois.

## A  Twisted BF theory

In this appendix, we review the topological quantum field theory (TQFT) for twisted BF theory [18, 78], given by the action,

$$S_{\text{TQFT}}[p, \tilde{a}, b, B] = \frac{N}{2\pi} \int b \wedge (\mathrm{d}\tilde{a} + B) + \frac{Np}{4\pi} \int b \wedge b + \frac{N}{2\pi} \int B \wedge \mathrm{d}\tilde{a}, \quad \text{(A.1)}$$

where $\tilde{a}_\mu$ and $\widetilde{\alpha}_\mu$ are dynamical $U(1)$ one-form gauge fields, $b_{\mu\nu}$ is a dynamical $U(1)$ two-form gauge field, and $B_{\mu\nu}$ is a background $U(1)$ two-form gauge field. The gauge field $\widetilde{\alpha}_\mu$ is a Lagrange multiplier that constrains $B_{\mu\nu}$ to be a $\mathbb{Z}_N$ gauge field. The parameters $N$ and $p$ are integers, and $p$ is defined mod $N$ on a spin manifold (mod $2N$ on a generic manifold). We first remove the background field, setting $B_{\mu\nu} = 0$. Consider the gauge transformation,

$$b \to b + \mathrm{d}\lambda, \qquad \tilde{a} \to \tilde{a} - p\,\lambda + \mathrm{d}\xi, \quad \text{(A.2)}$$

where $\xi$ is a $2\pi$ periodic scalar and $\lambda_\mu$ is a $U(1)$ one-form gauge field. If this TQFT is placed on a closed four-manifold, then the partition function is invariant under Eq. (A.2) if $N$ and $p$ are integers and $Np$ is even. If the manifold is a spin manifold, then $N$ and $p$ can be arbitrary integers.

We can form gauge invariant operators of the form,

$$\mathcal{D}(\gamma, \Omega)^K = \exp\left( iK \oint_\gamma \tilde{a} + ipK \int_\Omega b \right), \qu\text{(A.3)}$$

where $\gamma$ is a loop and $\Omega$ is an open surface bounding $\gamma$. The local equation of motion for $b_{\mu\nu}$ is that

$$\mathrm{d}\tilde{a} + p\,b = 0, \quad \text{(A.4)}$$

which renders $\mathcal{D}(\gamma, \Omega)^K$ trivial unless it is a genuine loop operator. Unlike loops attached to surfaces, genuine loops can be noncontractible and are not necessarily trivial globally. The surface in $\mathcal{D}(\gamma, \Omega)^K$ is trivial only if $pK \in N\mathbb{Z}$. This condition is satisfied by $K = \frac{N}{L}q$ where $L = \gcd(N, p)$ and $q$ is an integer mod $L$. Thus, there are $L$ physically distinct genuine loop operators, given by

$$\mathcal{T}(\gamma)^{Nq/L} = \exp\left( i\,\frac{Nq}{L} \oint_\gamma \tilde{a} \right), \qu\text{(A.5)}$$

where $\gamma$ can now be a non-contractible loop.

We can also form an operator supported on a closed surface $\Sigma$, given by

$$\mathcal{U}(\Sigma)^{q'} = \exp\left(-i\,q' \oint_\Sigma b\right). \tag{A.6}$$

Here, $q'$ is equivalent to $q' + N$ since $\tilde{a}_\mu$ is a Lagrange multiplier that constrains $b_{\mu\nu}$ to be a $\mathbb{Z}_N$ gauge field. However, $q'$ is also equivalent to $q' + p$ because of Eq. (A.4). Hence, $q'$ is defined mod $L$, and there are $L$ distinct surface operators.

The genuine loop operators and surface operators have correlation functions given by

$$\langle \mathcal{T}(\gamma)^{Nq/L} \mathcal{U}(\Sigma)^{q'} \rangle = \exp\left(\frac{2\pi i}{L} q\,q'\,\Phi(\gamma, \Sigma)\right), \tag{A.7}$$

where $\Phi(\gamma, \Sigma)$ is the linking number of $\gamma$ and $\Sigma$ in (3+1)d spacetime. Thus, the topological order realized by the loop and surface operators is equivalent to a topological $\mathbb{Z}_L$ gauge theory. A key distinction, however, is that $\mathcal{T}(\gamma)^{N/L}$ can represent the worldline of a fermionic particle if $Np/L^2$ is odd.

The TQFT has a $\mathbb{Z}_N$ one-form symmetry that acts as

$$\tilde{a} \to \tilde{a} + \eta, \tag{A.8}$$

where $\eta$ is a flat connection, satisfying $\mathrm{d}\eta = 0$ locally and $\oint \eta \in (2\pi/N)\mathbb{Z}$ globally. The surface operator that acts with this one-form symmetry is $\mathcal{U}(\Sigma)$, and the background field $B_{\mu\nu}$ probes this symmetry. The loop operators $\mathcal{T}(\gamma)^{N/L}$ transform nontrivially under this global symmetry, so the $\mathbb{Z}_N$ one-form symmetry is spontaneously broken to $\mathbb{Z}_{N/L}$ at low energies.

Next, we reintroduce the background field $B_{\mu\nu}$, which will allow us to observe consequences of the unbroken $\mathbb{Z}_{N/L}$ one-form symmetry. The theory is invariant under gauge transformations,

$$b \to b + \mathrm{d}\lambda, \qquad \tilde{a} \to \tilde{a} - p\,\lambda + \mathrm{d}\xi - \tilde{\lambda}, \qquad B \to B + \mathrm{d}\tilde{\lambda}, \qquad \widetilde{\alpha} \to \widetilde{\alpha} - \lambda + \mathrm{d}\tilde{\xi}, \tag{A.9}$$

where $\xi$ and $\tilde{\xi}$ are $2\pi$ periodic scalar fields while $\lambda_\mu$ and $\tilde{\lambda}_\mu$ are $U(1)$ one-form gauge fields. Integrating out $\tilde{a}_\mu$ and $\widetilde{\alpha}_\mu$ respectively constrain $b_{\mu\nu}$ and $B_{\mu\nu}$ to be $\mathbb{Z}_N$ gauge fields, so they are locally trivial, but globally they satisfy

$$\oint_\Sigma b = \frac{2\pi\ell}{N}, \qquad \oint_\Sigma B \in \frac{2\pi\tilde{\ell}}{N}, \qquad \ell, \tilde{\ell} \in \mathbb{Z}, \tag{A.10}$$

for any closed surface $\Sigma$. The action that remains after integrating out the Lagrange multipliers is

$$S_{\mathrm{eff}}[B] = \frac{N}{2\pi} \int b \wedge B + \frac{Np}{4\pi} \int b \wedge b. \tag{A.11}$$

We next integrate out $b_{\mu\nu}$. Both $b_{\mu\nu}$ and $B_{\mu\nu}$ are locally trivial, so we only have the global equation of motion,

$$\frac{2\pi}{N}\left(\tilde{\ell} + p\,\ell\right) = \left(\oint_\Sigma B + p \oint_\Sigma b\right) \in 2\pi\mathbb{Z}, \tag{A.12}$$

for any closed surface $\Sigma$. We thus find that

$$\left(\tilde{\ell} + p\,\ell\right) \in N\mathbb{Z}, \tag{A.13}$$

which is consistent only if $\tilde{\ell} \in L\mathbb{Z}$, where $L = \gcd(N, p)$. Hence, we must have

$$\oint_\Sigma B \in \frac{2\pi}{N/L}\mathbb{Z}, \tag{A.14}$$

which implies that $B_{\mu\nu}$ must be probing the unbroken $\mathbb{Z}_{N/L}$ subgroup of the $\mathbb{Z}_N$ one-form symmetry. This restriction of the background field to configurations consistent with the unbroken symmetry is analogous to the Meissner effect in a superconductor. To integrate out $b_{\mu\nu}$, we must solve Eq. (A.12) for $\ell$. Writing $\tilde{\ell} = L\ell'$, where $\ell' \in \mathbb{Z}$, a solution is

$$\frac{N}{2\pi} \oint_\Sigma b = \ell = r\,\ell' = \frac{rN}{2\pi L} \oint_\Sigma B \quad \text{mod } \frac{N}{L}\,, \tag{A.15}$$

where $r$ is an integer such that $r\,p = -L \bmod N$. (Such an $r$ must always exist.) The response after integrating out all dynamical fields is then

$$S_{\text{resp}}[B] = \frac{\frac{N}{L}r}{4\pi} \int B \wedge B\,, \tag{A.16}$$

signaling nontrivial SPT order for the unbroken $\mathbb{Z}_{N/L}$ one-form symmetry.

## B  Charge conjugation and Majorana condition

We briefly review Majorana fermions in this appendix. A more detailed recent discussion may be found in Ref. [147]. For the $\eta_{\mu\nu} = \text{diag}(1,-1,-1,-1)$ metric, the Dirac equation admits real spinor solutions if the Dirac matrices are purely imaginary, which is true in the Majorana basis,

$$\gamma_M^0 = \begin{pmatrix} 0 & \sigma^2 \\ \sigma^2 & 0 \end{pmatrix}, \qquad \gamma_M^1 = \begin{pmatrix} i\sigma^3 & 0 \\ 0 & i\sigma^3 \end{pmatrix}, \qquad \gamma_M^2 = \begin{pmatrix} 0 & -\sigma^2 \\ \sigma^2 & 0 \end{pmatrix},$$
$$\gamma_M^3 = \begin{pmatrix} -i\sigma^1 & 0 \\ 0 & -i\sigma^1 \end{pmatrix}, \qquad \gamma_M^5 = \begin{pmatrix} \sigma^2 & 0 \\ 0 & -\sigma^2 \end{pmatrix}. \tag{B.1}$$

The Dirac matrices $\gamma^\mu$ in a generic basis are related to the above $\gamma_M^\mu$ by a similarity transformation $M$ as

$$M\gamma^\mu M^{-1} = \gamma_M^\mu\,. \tag{B.2}$$

Because the $\gamma_M^\mu$ are purely imaginary, there must be a matrix $U_C$ such that

$$U_C\,\gamma^\mu\,U_C^{-1} = -(\gamma^\mu)^*\,, \tag{B.3}$$

where $U_C = (M^*)^{-1}M$.

In the Majorana basis, a Majorana fermion $\psi_M$ obeys the constraint,

$$(\psi_M^\dagger)^T = \psi_M\,. \tag{B.4}$$

To determine the analogue of this condition in a generic basis, we express $\psi_M = M\psi$ to obtain the condition,

$$(\psi^\dagger)^T = U_C\,\psi\,. \tag{B.5}$$

In terms of the matrix $\mathcal{C} = (U_C)^T\gamma^0$, the Majorana condition on $\psi$ is

$$\bar{\psi} = \psi^\dagger\gamma^0 = \psi^T(U_C)^T\gamma^0 = \psi^T\mathcal{C}\,. \tag{B.6}$$

This version of the Majorana condition is what generalizes most naturally to other spacetime metrics. For Dirac matrices in any basis, it can be shown [147] that there exists a unitary matrix $\mathcal{C}$ that satisfies

$$\mathcal{C}\gamma^\mu\mathcal{C}^{-1} = -(\gamma^\mu)^T\,, \qquad \mathcal{C}^T = -\mathcal{C}\,, \tag{B.7}$$

for both sign conventions of Lorentzian signature and for Euclidean signature.

As an example, we can take $\mathcal{C} = \gamma^0$ in the Majorana basis. Similarly, if we use the Weyl basis for Dirac matrices,

$$\gamma^\mu = \begin{pmatrix} 0 & \sigma^\mu \\ \bar{\sigma}^\mu & 0 \end{pmatrix}, \qquad \gamma^5 = i\gamma^0\gamma^1\gamma^2\gamma^3 = \begin{pmatrix} -\mathbb{I}_2 & 0 \\ 0 & \mathbb{I}_2 \end{pmatrix}, \tag{B.8}$$

where $\sigma^\mu = (\mathbb{I}_2, \sigma^j)$ and $\bar{\sigma}^\mu = (\mathbb{I}_2, -\sigma^j)$, then

$$\mathcal{C} = i\gamma^0\gamma^2 = \begin{pmatrix} -i\sigma^2 & 0 \\ 0 & i\sigma^2 \end{pmatrix}, \tag{B.9}$$

is a suitable charge conjugation matrix.

The charge conjugation matrix $\mathcal{C}$ may be used to define the action for a Majorana fermion in an arbitrary basis for the Dirac matrices. The Lagrangian density for a single Majorana fermion of mass $m$ is

$$\mathcal{L}_M = \frac{1}{2}\psi^T(x)\mathcal{C}(i\not{\partial} - m)\psi(x). \tag{B.10}$$

The partition function is

$$Z_M = \int \mathcal{D}\psi \exp\left(i\int d^4x \, \mathcal{L}_M\right) = \mathrm{Pf}[\mathcal{C}(i\not{\partial} - m)], \tag{B.11}$$

where Pf denotes the Pfaffian.

# C  Thermal response for fermions

Here, we summarize our conventions for curved spacetime and review how to couple fermions to a background metric $g_{\mu\nu}$, which is useful for keeping track of thermal response [88–91]. The spacetime metric $g_{\mu\nu}$ can be expressed in terms of a set of local Lorentz frame fields $e_\mu{}^a$ as

$$g_{\mu\nu} = e_\mu{}^a e_\nu{}^b \eta_{ab}, \tag{C.1}$$

where $\eta_{ab}$ is the Minkowski metric. The Christoffel symbols are constructed from the metric as

$$\Gamma^\mu{}_{\nu\lambda} = \frac{1}{2}g^{\mu\sigma}\left(\partial_\nu g_{\lambda\sigma} + \partial_\lambda g_{\nu\sigma} - \partial_\sigma g_{\nu\lambda}\right). \tag{C.2}$$

The spin connection is

$$\omega_\mu{}^{ab} = e_\nu{}^a\left(\partial_\mu e^{\nu b} + \Gamma^\nu{}_{\mu\lambda}e^{\lambda b}\right), \tag{C.3}$$

which can be used to define a one-form,

$$\omega^{ab} = \omega_\mu{}^{ab}\,dx^\mu, \tag{C.4}$$

and a corresponding curvature two-form,

$$R^a{}_b = d\omega^a{}_b + \omega^a{}_c \wedge \omega^c{}_b. \tag{C.5}$$

Using the notation $\mathrm{Tr}(R \wedge R) = R^a{}_b \wedge R^b{}_a$, the gravitational theta term is

$$S_g = \frac{\theta_g}{384\pi^2}\int \mathrm{Tr}(R \wedge R). \tag{C.6}$$

This term is quantized so that $\theta_g$ has periodicity $2\pi$ on a spin manifold and periodicity $32\pi$ on a generic four-manifold. This bulk response leads to a boundary gravitational Chern-Simons response with chiral central charge $c = \theta_g/4\pi$. The boundary thermal Hall conductivity is then

$$\kappa_{xy} = c \, \frac{\pi k_B^2 T}{6\hbar}, \tag{C.7}$$

where $T$ is the temperature.

Next, we review how to place fermionic fields in curved spacetime. The Dirac matrices in flat spacetime $\gamma^a$ satisfy

$$\{\gamma_a, \gamma_b\} = 2\, \eta_{ab}. \tag{C.8}$$

The generators of $\mathrm{Spin}(1,3)$, the double cover of the Lorentz group, are

$$\sigma_{ab} = \frac{i}{4}[\gamma_a, \gamma_b], \tag{C.9}$$

and obey the algebra,

$$[\sigma_{ab}, \sigma_{cd}] = i\left(g_{ad}\,\sigma_{bc} + g_{bc}\,\sigma_{ad} - g_{ac}\,\sigma_{bd} - g_{bd}\,\sigma_{ac}\right). \tag{C.10}$$

To couple a Majorana fermion $\psi(x)$ to a background metric $g_{\mu\nu}(x)$, we introduce the covariant derivative,

$$\nabla_\mu \psi = \left(\partial_\mu - \frac{i}{2}\,\omega_\mu{}^{ab}\,\sigma_{ab}\right)\psi. \tag{C.11}$$

Additionally, we define

$$\slashed{\nabla} = e^\mu{}_a(x)\,\gamma^a\,\nabla_\mu. \tag{C.12}$$

The action of the fermion coupled to the metric is

$$S = \int d^4x \sqrt{-g}\, \frac{1}{2}\, \psi^T(x)\, \mathcal{C}\, (i\slashed{\nabla} - m)\, \psi(x), \tag{C.13}$$

where $g$ is the determinant of $g_{\mu\nu}$ and $m$ is the fermion mass.

# D  Topological transition on the lattice

Here, we provide a lattice gauge theory model in the Hamiltonian formalism [148] that realizes the topological phase transition discussed in Section 4. See Refs. [149–151] for a review of lattice gauge theory.[18] First, we review how to place $PSU(N) = SU(N)/\mathbb{Z}_N$ gauge theory on the lattice [78,152,153] by gauging the $\mathbb{Z}_N$ electric one-form symmetry of $SU(N)$ lattice gauge theory.

We work on a cubic lattice. On each link $(\mathbf{r}; j)$, which is labeled by a lattice point $\mathbf{r}$ and a direction $j$, we place an $SU(N)$ gauge variable $U_j(\mathbf{r}) = e^{i T^\alpha A_j^\alpha(\mathbf{r})}$, where $T^\alpha$ are the generators of the $\mathfrak{su}(N)$ Lie algebra in the fundamental representation. On each plaquette $(\mathbf{r}; j, k)$, we define the operator,

$$W_{jk}(\mathbf{r}) = U_j(\mathbf{r})\, U_k(\mathbf{r} + \mathbf{e}_j)\, U_j^{-1}(\mathbf{r} + \mathbf{e}_k)\, U_k^{-1}(\mathbf{r}), \tag{D.1}$$

where $\mathbf{e}_j$ is a unit vector in the $j$ direction (and we set the lattice constant to unity). The Hamiltonian for $SU(N)$ lattice gauge theory is

$$H_{SU(N)} = \frac{g^2}{2}\sum_{\mathbf{r};j} E_j^\alpha(\mathbf{r})\, E_j^\alpha(\mathbf{r}) - \frac{1}{2g^2}\sum_{\mathbf{r};j,k} \mathrm{Tr}\left(W_{jk}(\mathbf{r}) + W_{jk}^\dagger(\mathbf{r})\right), \tag{D.2}$$

---

[18]We note, however, that the Gauss law for $SU(2)$ lattice gauge theory is incorrect in Ref. [151]. The author of this (otherwise excellent) textbook is aware of this issue.

where the self-adjoint operators $E_j^\alpha(\mathbf{r})$ are defined on links and obey

$$[A_j^\alpha(\mathbf{r}), E_k^\beta(\mathbf{r}')] = i\,\delta_{\mathbf{r},\mathbf{r}'}\,\delta_{j,k}\,\delta^{\alpha,\beta}\,, \qquad [E_j^\alpha(\mathbf{r}), E_k^\beta(\mathbf{r}')] = i\,f^{\alpha\beta\gamma}\,E_j^\gamma(\mathbf{r})\,\delta_{\mathbf{r},\mathbf{r}'}\,\delta_{j,k}\,, \qquad \text{(D.3)}$$

which also implies

$$[E_j^\alpha(\mathbf{r}), U_k(\mathbf{r}')] = T^\alpha\,U_j(\mathbf{r})\,\delta_{\mathbf{r},\mathbf{r}'}\,\delta_{j,k}\,,$$
$$e^{i\,\xi^\alpha E_j^\alpha(\mathbf{r})}\,U_k(\mathbf{r}')\,e^{-i\,\xi^\beta E_j^\beta(\mathbf{r})} = \exp\left(i\,\xi^\alpha\,T^\alpha\,\delta_{\mathbf{r},\mathbf{r}'}\,\delta_{j,k}\right)U_k(\mathbf{r}')\,, \qquad \text{(D.4)}$$

for some parameters $\xi^\alpha$. The generators of $SU(N)$ gauge transformations are

$$Q^\alpha(\mathbf{r}) = \sum_{j=1}^{3}\left[E_j^\alpha(\mathbf{r}) - (U_j^{\mathrm{adj}})^{\beta\alpha}(\mathbf{r}-\mathbf{e}_j)\,E_j^\beta(\mathbf{r}-\mathbf{e}_j)\right], \qquad \text{(D.5)}$$

where $(U_j^{\mathrm{adj}})^{\alpha\beta}(\mathbf{r}) = e^{i\,(T_{\mathrm{adj}}^\gamma)^{\alpha\beta}A_j^\gamma(\mathbf{r})} = e^{f^{\alpha\beta\gamma}A_j^\gamma(\mathbf{r})}$ are matrix elements of the gauge variables in the adjoint representation. The $Q^\alpha(\mathbf{r})$ commute with the Hamiltonian, $[Q^\alpha(\mathbf{r}), H_{SU(N)}] = 0$, and annihilate physical states so that $Q^\alpha(\mathbf{r})|\text{phys}\rangle = 0$.

To obtain $PSU(N)$ lattice gauge theory, we now gauge the $\mathbb{Z}_N$ electric one-form symmetry of $SU(N)$ lattice gauge theory. To couple to a $\mathbb{Z}_N$ two-form gauge field, we introduce unitary operators $\sigma_{jk}(\mathbf{r})$ and $\tau_{jk}(\mathbf{r})$ on plaquettes that satisfy

$$[\sigma_{jk}(\mathbf{r})]^N = [\tau_{jk}(\mathbf{r})]^N = 1\,,$$
$$\sigma_{jk}(\mathbf{r})\,\tau_{j'k'}(\mathbf{r}') = \exp\left(\frac{2\pi i}{N}\,\delta_{\mathbf{r},\mathbf{r}'}\,\delta_{j,j'}\,\delta_{k,k'}\right)\tau_{j'k'}(\mathbf{r}')\,\sigma_{jk}(\mathbf{r})\,. \qquad \text{(D.6)}$$

In addition, we have $\tau_{kj}(\mathbf{r}) = \tau_{jk}^\dagger(\mathbf{r})$ and $\sigma_{kj}(\mathbf{r}) = \sigma_{jk}^\dagger(\mathbf{r})$. The three-form analogue of a field strength, defined on cubes $(\mathbf{r}; 1, 2, 3)$, is

$$h_{123}(\mathbf{r}) = \sigma_{12}(\mathbf{r})\,\sigma_{12}^\dagger(\mathbf{r}+\mathbf{e}_3)\,\sigma_{23}(\mathbf{r})\,\sigma_{23}^\dagger(\mathbf{r}+\mathbf{e}_1)\,\sigma_{31}(\mathbf{r})\,\sigma_{31}^\dagger(\mathbf{r}+\mathbf{e}_2)\,. \qquad \text{(D.7)}$$

The Hamiltonian for $PSU(N)$ lattice gauge theory is

$$\begin{aligned}
H_{PSU(N)} = {}&\frac{g^2}{2}\sum_{\mathbf{r};j}E_j^\alpha(\mathbf{r})E_j^\alpha(\mathbf{r}) - \frac{1}{2g^2}\sum_{\mathbf{r};j,k}\mathrm{Tr}\left(W_{jk}(\mathbf{r})\,\sigma_{jk}(\mathbf{r}) + W_{jk}^\dagger(\mathbf{r})\,\sigma_{jk}^\dagger(\mathbf{r})\right) \\
&- \frac{1}{2\tilde{g}^2}\sum_{\mathbf{r}}\left(h_{123}(\mathbf{r}) + h_{123}^\dagger(\mathbf{r})\right) - \frac{1}{2}\sum_{\mathbf{r};j,k}\left(\tau_{jk}(\mathbf{r}) + \tau_{jk}^\dagger(\mathbf{r})\right).
\end{aligned} \qquad \text{(D.8)}$$

The $\mathbb{Z}_N$ two-form gauge field has its own Gauss law. To define the operator for this gauge charge, we must identify the operator that generates the $\mathbb{Z}_N$ center transformation of $SU(N)$ on each link. We can always take a generator of $SU(N)$ to be diagonal, which we take to be

$$T^{\alpha_0} = \sqrt{\frac{N}{2(N-1)}}\,\mathrm{diag}\left(\frac{1}{N}, \frac{1}{N}, \ldots, \frac{1}{N}, -1+\frac{1}{N}\right), \qquad \text{(D.9)}$$

for some $\alpha_0$. A $\mathbb{Z}_N$ center transformation on $U_k(\mathbf{r})$ is then given by

$$e^{i\,\eta E_j^{\alpha_0}(\mathbf{r})}\,U_k(\mathbf{r}')\,e^{-i\,\eta E_j^{\alpha_0}(\mathbf{r})} = e^{2\pi i\,\delta_{j,k}\,\delta_{\mathbf{r},\mathbf{r}'}/N}\,U_k(\mathbf{r}')\,, \qquad \text{(D.10)}$$

where $\eta = 2\pi\sqrt{2(N-1)/N}$. For each link $(\mathbf{r}, j)$, the generator of gauge transformations for the two-form gauge field is

$$\mathcal{Q}_j(\mathbf{r}) = e^{i\,\eta E_j^{\alpha_0}(\mathbf{r})}\prod_{k\neq j}\tau_{jk}(\mathbf{r})\,\tau_{jk}^\dagger(\mathbf{r}-\mathbf{e}_k)\,. \qquad \text{(D.11)}$$

This operator commutes with the Hamiltonian, $[\mathcal{Q}_j(\mathbf{r}), H_{PSU(N)}] = 0$, and physical states must be invariant under this operator, $\mathcal{Q}_j(\mathbf{r})|\text{phys}\rangle = |\text{phys}\rangle$. This constraint also ensures that the 't Hooft loop, which must be attached to a surface in $SU(N)$ gauge theory, becomes a genuine loop operator in $PSU(N)$ gauge theory. Indeed, the 't Hooft loop in $SU(N)$ gauge theory, which is defined on a loop $\tilde{\gamma}$ on the dual lattice attached to a surface $\widetilde{\Sigma}$, is

$$\mathcal{T}(\tilde{\gamma}) = \prod_{(\mathbf{r};j) \in \widetilde{\Sigma}} e^{-i\eta E_j^{\alpha_0}(\mathbf{r})}, \tag{D.12}$$

where the product is over links intersecting $\widetilde{\Sigma}$. However, the invariance of states under $\mathcal{Q}_j(\mathbf{r})$ in $PSU(N)$ gauge theory ensures that the choice of surface $\widetilde{\Sigma}$ does not matter since this operator acts on physical states in the same way as

$$\mathcal{T}(\tilde{\gamma}) = \prod_{(\mathbf{r};j,k) \in \tilde{\gamma}} \tau_{jk}(\mathbf{r}), \tag{D.13}$$

where the product is now over plaquettes intersecting the loop $\tilde{\gamma}$.

Before coupling the gauge fields to fermions, let us check that $\mathbb{Z}_N$ topological order is produced in a certain limit of the $PSU(N)$ lattice gauge theory. We take the limit $\tilde{g}^2 \to 0$ so that the terms in the Hamiltonian with $\tau_{jk}(\mathbf{r})$ and its Hermitian conjugate may be ignored. Now the ground state $|\psi_0\rangle$ must satisfy,

$$h_{123}(\mathbf{r})|\psi_0\rangle = h_{123}^\dagger(\mathbf{r})|\psi_0\rangle = |\psi_0\rangle, \tag{D.14}$$

so the 't Hooft loops cannot end. In this limit, there is an exact $\mathbb{Z}_N$ magnetic one-form global symmetry, as in the continuum $PSU(N)$ gauge theory. For finite $\tilde{g}^2$ (but not too large), we expect that this symmetry will be emergent. The operator that acts with the $\mathbb{Z}_N$ one-form symmetry, defined on a noncontractible surface $\Sigma$,

$$\mathcal{U}(\Sigma) = \prod_{(\mathbf{r};j,k) \in \Sigma} \sigma_{jk}(\mathbf{r}), \tag{D.15}$$

now commutes with the Hamiltonian. Taking $g^2 \to \infty$ also, the ground state must satisfy,

$$E_j^\alpha(\mathbf{r})|\psi_0\rangle = 0, \tag{D.16}$$

so that

$$\langle\psi_0|\mathcal{T}(\tilde{\gamma})|\psi_0\rangle = 1, \tag{D.17}$$

for any contractible loop $\tilde{\gamma}$ on the dual lattice. For finite $g$, the 't Hooft loop will instead have a perimeter law. Hence, the $\mathbb{Z}_N$ magnetic one-form symmetry will ultimately be spontaneously broken at low energies by deconfined monopoles, leading to $\mathbb{Z}_N$ topological order.

Another way to observe the topological order is to note that for $g^2 \to \infty$ the 't Hooft loop $\mathcal{T}(\tilde{\gamma})$ commutes with the Hamiltonian. For noncontractible $\tilde{\gamma}$ this operator generates a $\mathbb{Z}_N$ two-form global symmetry that acts on the surface operator $\mathcal{U}(\Sigma)$ for noncontractible $\Sigma$. For example, we take periodic boundary conditions in all directions. Let $\tilde{\gamma}$ be a noncontractible loop in the $j$ direction and $\Sigma$ be a plane perpendicular to the $j$ direction. The operators $\mathcal{U}(\Sigma)$ and $\mathcal{T}(\tilde{\gamma})$ both commute with the Hamiltonian and obey

$$\mathcal{U}(\Sigma)\mathcal{T}(\tilde{\gamma}) = e^{2\pi i/N}\mathcal{U}(\Sigma)\mathcal{T}(\tilde{\gamma}), \tag{D.18}$$

which implies a ground state degeneracy of $N^3$ on the torus, thus confirming the $\mathbb{Z}_N$ topological order.

Finally, we introduce fermions. On each site $\mathbf{r}$, we place a four-component Majorana spinor $\psi_{\mathfrak{a}}(\mathbf{r})$, which obeys the anticommutation relations,

$$\left\{\psi_{\mathfrak{a}}(\mathbf{r}), \psi_{\mathfrak{b}}(\mathbf{r}')\right\} = (\gamma^0 \mathcal{C}^{-1})_{\mathfrak{a}\mathfrak{b}} \, \delta_{\mathbf{r}, \mathbf{r}'}, \tag{D.19}$$

where $\mathfrak{a}$ and $\mathfrak{b}$ are spinor indices and $\mathcal{C}$ is the charge conjugation matrix (see Appendix B). The Hamiltonian for a single free Majorana fermion $\psi(\mathbf{r})$ of mass $m$ is

$$H_M = \sum_{\mathbf{r}} \frac{1}{2} \psi^T(\mathbf{r}) \mathcal{C} (D_W + m) \psi(\mathbf{r}), \tag{D.20}$$

where $\psi^T(\mathbf{r})$ is the transpose of the spinor $\psi(\mathbf{r})$ and $D_W$ is the Wilson operator [53],

$$D_W = \frac{1}{2} \sum_{j=1}^{3} \left[ -i \gamma^j \left(\Delta_j^+ + \Delta_j^-\right) - \left(\Delta_j^+ - \Delta_j^-\right) \right], \tag{D.21}$$

where $\Delta_j^+ \psi(\mathbf{r}) = \psi(\mathbf{r} + \mathbf{e}_j) - \psi(\mathbf{r})$ and $\Delta_j^- \psi(\mathbf{r}) = \psi(\mathbf{r}) - \psi(\mathbf{r} - \mathbf{e}_j)$. The gapped phases of Eq. (D.20) are time-reversal invariant topological superconductors in class DIII, characterized by a topological invariant $\nu$, which is classified by $\mathbb{Z}$ for free fermions [154–156] but collapses to $\mathbb{Z}_{16}$ when interactions are taken into account [85, 126–128]. The topological invariant for Eq. (D.20) as a function of $m$ is

$$\nu = \begin{cases} 0, & m < -6, \; m > 0, \\ -1, & -6 < m < -4, \; -2 < m < 0, \\ 2, & -4 < m < -2. \end{cases} \tag{D.22}$$

In particular, notice that the $m > 0$ phase is a trivial superconductor, and there is a transition at $m = 0$, where the Majorana fermion becomes massless, to the phase at $-2 < m < 0$, which is a topological superconductor with a gravitational response of $\theta_g = \pi$ (see Appendix C). Thus, in the continuum limit, the transition at $m = 0$ becomes the transition of a single Majorana fermion whose mass changes sign. If we take $N_f \, (N^2 - 1)$ copies of this Majorana fermion and couple them to a $PSU(N)$ gauge field in the adjoint representation, then for $N$ even and a sufficiently large odd number of flavors $N_f$, there will be a continuous transition at $m = 0$ that corresponds to the topological transition discussed for the continuum $PSU(N)$ adjoint QCD theory in Section 4.

To couple fermion fields $\psi(\mathbf{r})$ that transform in a representation $\mathcal{R}$ of the gauge field, we define

$$\begin{aligned} D_j^+(\mathcal{R})\psi(\mathbf{r}) &= U_j^{\mathcal{R}}(\mathbf{r})\psi(\mathbf{r}+\mathbf{e}_j) - \psi(\mathbf{r}), \\ D_j^-(\mathcal{R})\psi(\mathbf{r}) &= \psi(\mathbf{r}) - (U_j^{\mathcal{R}})^{-1}(\mathbf{r}-\mathbf{e}_j)\psi(\mathbf{r}-\mathbf{e}_j), \end{aligned} \tag{D.23}$$

where $U_j^{\mathcal{R}}(\mathbf{r}) = e^{i \, T_{\mathcal{R}}^\alpha A_j^\alpha(\mathbf{r})}$ are the gauge variables in representation $\mathcal{R}$. We also define

$$\mathcal{D}_W^{\mathcal{R}} = \frac{1}{2} \sum_{j=1}^{3} \left[ -i \gamma^j \left(D_j^+(\mathcal{R}) + D_j^-(\mathcal{R})\right) - \left(D_j^+(\mathcal{R}) - D_j^-(\mathcal{R})\right) \right]. \tag{D.24}$$

Taking $\mathcal{R}$ to be the adjoint representation and introducing $N_f$ flavors of Majorana fermions, indexed by $J$, the Hamiltonian is

$$H = \sum_{J=1}^{N_f} \sum_{\mathbf{r}} \frac{1}{2} \psi_J^T(r) \mathcal{C} (\mathcal{D}_W^{\text{adj}} + m) \psi_J(r) + H_{PSU(N)}, \tag{D.25}$$

where $H_{PSU(N)}$ is given in Eq. (D.8). Even with the coupling to fermions, the one-form gauge charge $\mathcal{Q}_j(\mathbf{r})$, defined in Eq. (D.11), remains the same. However, the generator of zero-form gauge transformations, Eq. (D.5), must be modified to

$$Q^{\alpha}(\mathbf{r}) = \sum_{j=1}^{3} \Big[ E_j^{\alpha}(\mathbf{r}) - (U_j^{\text{adj}})^{\beta\alpha}(\mathbf{r} - \mathbf{e}_j) E_j^{\beta}(\mathbf{r} - \mathbf{e}_j) \Big] - \frac{1}{2} \sum_{J=1}^{N_f} \psi_J^T(\mathbf{r}) \mathcal{C}\gamma^0 \, T_{\text{adj}}^{\alpha} \psi_J(\mathbf{r}). \quad \text{(D.26)}$$

This operator commutes with the Hamiltonian, Eq. (D.25). As discussed above, for $N$ even and sufficiently large odd $N_f$, there will be a continuous transition in the lattice Hamiltonian, Eq. (D.25), at $m = 0$. Taking $g^2$ large and $\tilde{g}^2$ small, the phase for $m > 0$ will be the $\mathbb{Z}_N$ topological order discussed in Section 4.2, and the $-2 < m < 0$ phase is the SET with $\mathbb{Z}_{N/2}$ topological order explained in Section 4.3.

As noted previously, for finite $\tilde{g}^2$ the $\mathbb{Z}_N$ magnetic one-form symmetry that is present in the continuum theory is explicitly broken by dynamical magnetic monopoles. This symmetry is expected to be emergent for small (but still finite) $\tilde{g}^2$. Because the magnetic monopoles in this lattice model, Eq. (D.25), transform trivially under the $SO(N_f)$ flavor symmetry and time-reversal symmetry, the SET orders arising from the Hamiltonian in Eq. (D.25) will have the trivial symmetry fractionalization class—the deconfined anyons will be bosonic Kramers singlets and $SO(N_f)$ tensors.

# E   Other gauge groups

In Section 4, we discuss a transition between SETs with different topological orders for $PSU(N)$ adjoint QCD. An analogue of this topological transition can occur for adjoint QCD with other gauge groups. Consider a gauge field with a gauge group $G_g$ coupled to $N_f$ odd flavors of Majorana fermions in the adjoint representation. As in Section 4, we impose an $SO(N_f)$ flavor symmetry and time-reversal symmetry, and we take $N_f$ large enough so that the transition is continuous. In the discussion below, $k$ will always be a nonnegative integer, and we regularize the theory so that the phase with positive fermion mass $m$ is the pure gauge theory with $\theta = 0$. The full magnetic one-form symmetry will be spontaneously broken in this phase. To determine the nature of the $m < 0$ phase, we use the relationship between the traditional $G_g$ theta term and discrete theta terms for various gauge groups as given in Refs. [50, 157].

If the gauge group is $G_g = \text{Sp}(4k+1)/\mathbb{Z}_2$, the magnetic one-form symmetry is $\mathbb{Z}_2$. The $m > 0$ phase has $\mathbb{Z}_2$ topological order, and at low energies, the $m < 0$ phase has a theta angle for $G_g$ of $\theta = \pi(4k+2)N_f$, leading to the effective action,

$$\begin{aligned}
S_{\text{eff}} = {} & \frac{2N_f(2k+1)(4k+1)}{4\pi} \int b \wedge b + \frac{(2k+1)(4k+1)\pi}{8\pi^2} \int \text{Tr}(F \wedge F) \\
& + \frac{N_f(2k+1)(4k+1)\pi}{384\pi^2} \int \text{Tr}(R \wedge R),
\end{aligned} \quad \text{(E.1)}$$

where $b_{\mu\nu}$ is a dynamical $\mathbb{Z}_2$ two-form gauge field, $F_{\mu\nu}$ is a background field for the $SO(N_f)$ flavor symmetry, and $R$ is the curvature two-form. Because $N_f(2k+1)(4k+1)$ is odd, there is no topological order in this phase. If we couple to a background field $B_{\mu\nu}$ for the $\mathbb{Z}_2$ magnetic one-form symmetry and integrate out $b_{\mu\nu}$, we find that this phase has nontrivial SPT order for the unbroken $\mathbb{Z}_2$ magnetic one-form symmetry. Given that $\text{Sp}(1) \cong SU(2)$, this analysis is consistent with our results in Section 4.

If the gauge group is $G_g = \mathrm{Spin}(8k+6)/\mathbb{Z}_4$, then there is a $\mathbb{Z}_4$ magnetic one-form symmetry, so the $m > 0$ phase has $\mathbb{Z}_4$ topological order. The $m < 0$ phase has a theta angle for $G_g$ of $\theta = \pi(8k+4)N_f$, giving the effective action,

$$
\begin{aligned}
S_{\mathrm{eff}} = {} & \frac{4(4k+3)(4k+2)N_f}{4\pi} \int b \wedge b + \frac{(4k+3)(8k+5)\pi}{8\pi^2} \int \mathrm{Tr}(F \wedge F) \\
& + \frac{N_f(4k+3)(8k+5)\pi}{384\pi^2} \int \mathrm{Tr}(R \wedge R),
\end{aligned}
\tag{E.2}
$$

where $b_{\mu\nu}$ is a dynamical $\mathbb{Z}_4$ two-form gauge field. Since $\gcd(4,(4k+3)(4k+2)N_f) = 2$, the $\mathbb{Z}_4$ one-form symmetry is spontaneously broken to $\mathbb{Z}_2$, and the unbroken $\mathbb{Z}_2$ one-form symmetry has nontrivial SPT order. For $\mathrm{Spin}(6) \cong SU(4)$, this conclusion is consistent with Section 4.

The gauge groups $\mathrm{Spin}(8k)/(\mathbb{Z}_2 \times \mathbb{Z}_2)$ and $\mathrm{Spin}(8k+4)/(\mathbb{Z}_2 \times \mathbb{Z}_2)$ have $\mathbb{Z}_2 \times \mathbb{Z}_2$ magnetic one-form symmetries. The $m > 0$ phase thus has $\mathbb{Z}_2 \times \mathbb{Z}_2$ topological order in both cases. If the gauge group is $G_g = \mathrm{Spin}(8k)/(\mathbb{Z}_2 \times \mathbb{Z}_2)$, then the $m < 0$ phase has a $G_g$ theta angle of $\theta = \pi(8k-2)N_f$, which leads to an effective action of

$$
S_{\mathrm{eff}} = \frac{2(4k-1)N_f}{2\pi} \int b_1 \wedge b_2 + \frac{4k(8k-1)\pi}{8\pi^2} \int \mathrm{Tr}(F \wedge F) + \frac{N_f 4k(8k-1)\pi}{384\pi^2} \int \mathrm{Tr}(R \wedge R), \tag{E.3}
$$

where $(b_1)_{\mu\nu}$ and $(b_2)_{\mu\nu}$ are dynamical $\mathbb{Z}_2$ two-form gauge fields. Since $(4k-1)N_f$ is odd, there is no topological order in this phase. If we couple to background fields, $(B_1)_{\mu\nu}$ and $(B_2)_{\mu\nu}$, for the two $\mathbb{Z}_2$ magnetic one-form symmetries and integrate out $(b_1)_{\mu\nu}$ and $(b_2)_{\mu\nu}$, we find that the $\mathbb{Z}_2$ magnetic one-form symmetries have a mixed SPT response,

$$
S_{\mathrm{mixed}}[B_1, B_2] = \frac{2}{2\pi} \int B_1 \wedge B_2, \tag{E.4}
$$

in addition to the zero-form SPT response in Eq. (E.3).

For $G_g = \mathrm{Spin}(8k+4)/(\mathbb{Z}_2 \times \mathbb{Z}_2)$, the $m < 0$ phase has a $G_g$ theta angle of $\theta = \pi(8k+2)N_f$ so that the effective action is

$$
\begin{aligned}
S_{\mathrm{eff}} = {} & \frac{2(4k+1)(2k+1)N_f}{4\pi} \int (b_1 \wedge b_1 + b_2 \wedge b_2) + \frac{(4k+2)(8k+3)\pi}{8\pi^2} \int \mathrm{Tr}(F \wedge F) \\
& + \frac{N_f(4k+2)(8k+3)\pi}{384\pi^2} \int \mathrm{Tr}(R \wedge R).
\end{aligned}
\tag{E.5}
$$

Since $(4k+1)(2k+1)N_f$ is odd, this phase is also an SPT. However, it has a different response for the magnetic one-form symmetry, given by

$$
S_{\mathrm{unmixed}}[B_1, B_2] = \frac{2}{4\pi} \int (B_1 \wedge B_1 + B_2 \wedge B_2), \tag{E.6}
$$

which does not couple the two background fields.

## F  Anomaly of non-invertible time-reversal

Here, we identify the conditions under which the non-invertible time-reversal symmetry associated with the operator $\mathsf{T}_n$, defined in Section 5.1, has a mixed anomaly with the $\mathbb{Z}_N$ one-form symmetry. Specifically, the anomaly implies that no trivially gapped phase can simultaneously preserve both symmetries.

We first consider the most general fermionic $\mathbb{Z}_N$ one-form SPT in (3+1)d, given by the action

$$S_{\text{SPT}}[B] = \frac{Np}{4\pi} \int B \wedge B \,, \tag{F.1}$$

where $p$ is an integer mod $N$ and $B_{\mu\nu}$ is a background $\mathbb{Z}_N$ two-form gauge field. The action for this SPT transforms under the non-invertible time-reversal transformation $\mathbf{K}_n$ to

$$S_{\mathbf{K}_n} = -\frac{Np}{4\pi} \int b \wedge b + \frac{N}{2\pi} \int b \wedge \beta + \frac{N(N-1)n}{4\pi} \int \beta \wedge \beta - \frac{N}{2\pi} \int \beta \wedge B \,, \tag{F.2}$$

where $b_{\mu\nu}$ and $\beta_{\mu\nu}$ are dynamical $\mathbb{Z}_N$ two-form gauge fields. To integrate out $b_{\mu\nu}$, we must have $\gcd(N,p) = 1$. Integrating out $b_{\mu\nu}$ then gives the action

$$S_{\text{eff}} = \frac{N(\ell + (N-1)n)}{4\pi} \int \beta \wedge \beta - \frac{N}{2\pi} \int \beta \wedge B \,, \tag{F.3}$$

where $\ell$ is an integer such that $\ell p = 1 \mod N$. Finally, integrating out $\beta_{\mu\nu}$, which requires $\gcd(N, \ell - n) = 1$, results in the SPT,

$$\widetilde{S}_{\text{SPT}}[B] = -\frac{N\tilde{\ell}}{4\pi} \int B \wedge B \,, \tag{F.4}$$

where $\tilde{\ell}$ is an integer such that $\tilde{\ell}(\ell - n) = 1 \mod N$.

The original SPT state, Eq. (F.1), is then invariant under the non-invertible time-reversal transformation $\mathbf{K}_n$ if $\tilde{\ell} = -p \mod N$, which implies that

$$1 = \tilde{\ell}(\ell - n) = -p(\ell - n) = -1 + pn \mod N \,. \tag{F.5}$$

Hence, we must have

$$pn = 2 \mod N \,, \tag{F.6}$$

for some $p$, indicating that $\gcd(N,n)$ is either 1 or 2. If $N$ or $n$ is odd, a solution for $p$ exists only if $\gcd(N,n) = 1$. If $N$ and $n$ are both even, there exists a solution for $p$ only if $\gcd(N/2, n/2) = 1$. Thus, there is an anomaly for $\gcd(N,n) > 1$ if $N$ or $n$ is odd and for $\gcd(N/2, n/2) > 1$ if $N$ and $n$ are both even. In these cases, no trivially gapped state can preserve both the $\mathbb{Z}_N$ one-form symmetry and non-invertible time-reversal symmetry.

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
