# Peer review of "Generalized symmetry enriched criticality in (3+1)d"

_SciPost Physics, doi:SciPost Phys. 19, 145 (2025)_

## Round 1 · Referee Report · Anonymous (Referee 1) · 2025-10-8

Strengths

Sections 2 and 3 furnish a clear and self-contained introduction to the subject.

Weaknesses

1) The point of the lattice model section 4.5 is unclear. If the lattice analysis provides additional insight into the phase structure, the authors should make it explicit. Otherwise, I suggest moving it to the appendix.

2) The discussion on page 23 about the symmetry fractional class is rough and incomplete. The author seems to aim at emphasizing the role of global symmetry in the topological phases, and I believe it would be worthwhile to expand this part of the discussion. I would suggest the author specify the group cohomology and provide more details if possible.

Report

This paper studies two massive phases of $PSU(N)$ agjoint QCD and the transition between them. While some aspects of this subject have been discussed in previous works (e.g., 2007.05915 and related studies), this paper presents interesting new results with clear potential for follow-up investigations. I believe this paper meets SciPost's criteria, but revisions are necessary before acceptance. In addition to the general weaknesses above, I list several specific minor issues below.

On page 5: ''we emphasize that these quasiparticles need not be magnetic monopoles of the electromagnetic field'' This phrasing is somewhat misleading, as the term ''electromagnetic field'' almost exclusively refers to U(1) gauge fields.

Equation (2.3): I suggest the author provide a more detailed definition in terms of the singular boundary condition.

Equations (2.8) and (2.11): the author has used two different constants $\rho$ and $\tilde{\rho}$ without an explanation. I believe it is beneficial to note that the constants are generally schema-dependent.

Equation (4.19) and (5.25): I assume this is the leading result at the limit $g\to 0$. Can the author provide what parameter controls this expansion? Is the next-order correction at the order $O(g^2)$, $O(m/\Lambda)$, or something else?

On page 34 above equation (5.6): As far as I understand, $T$ and $D_n$ by themselves are not symmetries for theoreis with $\theta=\pi n$. Rather, they are topological interfaces between different theories. The author's statement "T is the standard invertible time-reversal operator" is slightly misleading. I suggest an improvement.

Recommendation

Ask for major revision

  • validity: -
  • significance: -
  • originality: -
  • clarity: -
  • formatting: -
  • grammar: -

Author:  Benjamin Moy  on 2025-10-20  [id 5945]

(in reply to Report 1 on 2025-10-08)

Thank you for your detailed and thoughtful comments. Below are replies to specific comments:

Referee's comment:

The point of the lattice model section 4.5 is unclear. If the lattice analysis provides additional insight into the phase structure, the authors should make it explicit. Otherwise, I suggest moving it to the appendix.

Reply:

I was independently considering whether to move the lattice model subsection to an appendix, so thank you for this constructive feedback. The primary utility of the lattice model is that it provides a natural way to study the phase structure upon explicitly breaking the $\mathbb{Z}_N$ magnetic one-form symmetry, which is more typical in condensed matter settings. Since this analysis was left to future work, I agree that it is better to move this subsection to the appendix and have done so in the updated version.

Referee's comment:

The discussion on page 23 about the symmetry fractional class is rough and incomplete. The author seems to aim at emphasizing the role of global symmetry in the topological phases, and I believe it would be worthwhile to expand this part of the discussion. I would suggest the author specify the group cohomology and provide more details if possible.

Reply:

Thanks for this suggestion. I have added more details to the discussion of symmetry fractionalization in Section 4.2 including the group cohomology. Correspondingly, I have added two paragraphs in Section 4.3 (on pp. 27-28) commenting on the role of the symmetry fractionalization in the other topologically ordered phase.

Referee's comment:

On page 5: ''we emphasize that these quasiparticles need not be magnetic monopoles of the electromagnetic field'' This phrasing is somewhat misleading, as the term ''electromagnetic field'' almost exclusively refers to U(1) gauge fields.

Reply:

I meant that the monopoles of the PSU(N) gauge theory should not be confused with monopoles of the U(1) electromagnetic gauge field. The paper has been edited to reflect that.

Referee's comment:

Equation (2.3): I suggest the author provide a more detailed definition in terms of the singular boundary condition.

Reply:

I have added a more explicit definition of the 't Hooft loop.

Referee's comment:

Equations (2.8) and (2.11): the author has used two different constants $\rho$ and $\tilde{\rho}$ without an explanation. I believe it is beneficial to note that the constants are generally schema-dependent.

Reply:

I have added a note that the constants depend on scheme.

Referee's comment:

Equation (4.19) and (5.25): I assume this is the leading result at the limit $g\to 0$. Can the author provide what parameter controls this expansion? Is the next-order correction at the order $O(g^2)$, $O(m/\Lambda)$ , or something else?

Reply:

The expansion parameter is $|m|/\Lambda_{\mathrm{UV}}$. To obtain higher order corrections, one should solve the coupled RG equations for $g$ and $m$ to lowest order in both parameters. This was done numerically in References 128 and 129, and their results agree with Eq. 4.19 for $|m|/\Lambda_\mathrm{UV}\ll 1$. Because $g$ is marginally irrelevant at the critical point, I expect that the next order correction to Eq. 4.19 should introduce logarithmic corrections to scaling, like for the Ising critical point in (3+1)d.

Referee's comment:

On page 34 above equation (5.6): As far as I understand, $T$ and $D_n$ by themselves are not symmetries for theoreis with $\theta=\pi n$. Rather, they are topological interfaces between different theories. The author's statement "T is the standard invertible time-reversal operator" is slightly misleading. I suggest an improvement.

Reply:

Thanks for catching this. I revised the statement, now referring to $\mathsf{T}$ as an interface that reverses orientation.

---

## Round 1 · Referee Report · Anonymous (Referee 2) · 2025-10-9

Report

The paper constructs and analyzes two classes of continuous quantum phase transitions in (3+1) dimensions, between phases that break different generalized global symmetries (in particular, one-form symmetries and non‐invertible symmetries). The authors focus on SU(N)/Z_N​ gauge theory coupled to Nf​ flavors of Majorana fermions in the adjoint representation. They argue that, for certain even N and sufficiently large odd Nf​, with time‐reversal and SO(Nf​) flavor symmetry, the massless theory can realize a quantum critical point between: A phase in which a ZN​ one-form symmetry is fully broken and a phase in which it is broken only down to ZN/2​ topological order. They also provide a lattice model displaying this transition, and analyze symmetry enhancements at the critical point, including non-invertible time-reversal analogues and show an analogue of deconfined quantum criticality for generalized symmetries.

The topic is timely. The interplay of generalized (higher-form / non-invertible) symmetries, topological order, and critical phenomena is at the frontier of condensed matter / high-energy duality lines of research. The paper finds interesting new examples of critical points enriched by generalized symmetries.

I believe this is a compelling and high-quality contribution to the study of generalized symmetries and critical phenomena. I recommend acceptance after minor revisions. Maybe it would be helpful that the author more more explicitly situate the results relative to prior work on deconfined quantum criticality, generalized symmetries in critical phases, and non-invertible symmetry studies. Perhaps a table or comparative discussion would help.

Recommendation

Publish (meets expectations and criteria for this Journal)

  • validity: good
  • significance: good
  • originality: good
  • clarity: good
  • formatting: good
  • grammar: good

Author:  Benjamin Moy  on 2025-10-20  [id 5944]

(in reply to Report 2 on 2025-10-09)

Thank you for your positive feedback and your suggestion. I have added a paragraph to the introduction at the bottom of p. 6 discussing previous work on critical phases and critical points for which non-invertible symmetry plays an important role.

---

## Round 2 · Referee Report · Anonymous (Referee 1) · 2025-10-29

Report

I would like to thank the author for the detailed response and improvements on the manuscript. I believe this paper meets the journal’s criteria for publication, and I recommend acceptance in its present form.

Recommendation

Publish (easily meets expectations and criteria for this Journal; among top 50%)

---

## Round 2 · Author Response

I thank both referees for their time, comments, and questions. I have responded to specific comments/questions from the referees using the "Reply to the above report" option. Note that all numbers for equations, sections, and references below and in my replies to the referees correspond to those in the more recent version of the manuscript.

---

## Round 2 · List of Changes

References added: 40, 42-48, 77, 78, 81, 112-120, 126, 127, 154

p. 5: “we emphasize that these quasiparticles need not be magnetic monopoles of the electromagnetic field” -> “we emphasize that these quasiparticles should not be confused with magnetic monopoles of the $U(1)$ electromagnetic field”

Just below Eq. 2.3: Added an explicit definition of the ’t Hooft loop

Just below Eq. 2.10 and just below Eq. 2.13: Added a comment that the constants associated with the perimeter law are scheme-dependent

p. 17: Moved a comment about Pin$^+$ structure from a footnote to the main text since it is relevant for the revised discussion of symmetry fractionalization

pp. 23-24: Added more details on symmetry fractionalization to Section 4.2

pp. 27-28: Added two paragraphs on how the SPT response for the SET phase studied in Section 4.3 is correlated with the symmetry fractionalization of SET studied in Section 4.2. (Figure 2 caption and Section 4.3 summary paragraph also updated accordingly.)

p. 30: Added a sentence after Eq. 4.19 about its regime of validity

p. 32: Instead of referring to T as the “standard invertible time-reversal operator”, call it an interface that reverses orientation

Moved lattice model subsection (previously Subsection 4.4) to a new appendix (Appendix D)

---

## Editorial Decision

published